# Continuum modeling of bioclogging of soil aquifer treatment systems segregating active and inactive biomass

Edwin Y. Saavedra Cifuentes[1], Alex Furman[2], Ravid Rosenzweig[3, 4], and Aaron I. Packman[1]

[1]Northwestern University, Evanston IL, USA
[2]Technion - Israel Institute of Technology, Haifa, Israel
[3]Geological Survey of Israel, Jerusalem, Israel
[4]Department of Geological and Environmental Sciences, Ben-Gurion University, Beer-Sheva , Israel

**Correspondence:** Edwin Saavedra Cifuentes (esaavedrac@u.northwestern.edu)

**Abstract.** Soil aquifer treatment (SAT) systems are used to remove pollutants from treated wastewater and store freshwater for reclamation and reuse. However, the accumulation of microbial biomass in the soil pore space, bioclogging, reduces water infiltration and hinders SAT efficiency. Since SAT systems play a crucial role in maintaining water resilience by providing an alternative to freshwater supply, optimizing their operation is essential to ensure their effectiveness. However, SAT systems are
complex and dynamic systems that involve coupled interactions between microbial activity, water infiltration, and bioclogging in unsaturated media. This work proposes a continuum model that accounts for all these processes while distinguishing between active and inactive biomass, with the latter split into labile and recalcitrant fractions. The model is used to replicate a laboratory column experiment of bioclogging under unsaturated conditions and to explore how to optimize the operation of SAT systems. Specifically, we determined optimal wetting and drying periods that maximize water input to the SAT system while maintaining
nutrient transformation rates. Our simulations show that the dry/wet time ratio controls biomass spatial distribution over depth. In contrast, the dry time extent dictates the degree of recovery of the soil relative to its initial (clean) infiltration capacity. We discuss the potential of this model to be extended to larger-scale experiments and to inform daily SAT operations in the field.

## 1 Introduction

Managed aquifer recharge (MAR) is the process of recharging water into an aquifer for later recovery (Dillon et al., 2019).
Soil-Aquifer Treatment (SAT) systems are a subset of MAR, where treated wastewater is infiltrated through the vadose zone to remove pollutants such as nutrients and pathogens (Bouwer, 1991; Fox et al., 2001; Sharma and Kennedy, 2017). MAR strategies and SAT systems are crucial for water resilience, providing an alternative to freshwater supply (Dillon et al., 2020). Conventional wastewater treatment is responsible for significant energy consumption (Electric Power Research Institute Inc., 2013). Replacing tertiary treatment with SAT systems using infiltration ponds represents a low-energy demand alternative
(Goren et al., 2014; Sharma and Kennedy, 2017). Reclaimed water from SAT systems has been used extensively; for example, more than $160 \times 10^6 \, \mathrm{m}^3/\mathrm{yr}$ of wastewater reclamation plant effluent is used annually for crop irrigation in Israel (Arad et al., 2023). Figure 1 shows a schematic of SAT operation: the effluent of a wastewater reclamation plant is placed on an infiltration

pond, where it infiltrates the soil and its quality improves during its passage through the subsurface until reaching an extraction well.

Water quality is significantly improved in SAT systems because, during infiltration, naturally occurring processes take place: filtration of suspended solids and fine particles, contaminant adsorption to soil minerals (Bradford et al., 2013), and biogeo-chemical transformations (Rauch and Drewes, 2005; Mienis and Arye, 2018; Gharoon and Pagilla, 2021). Despite the large spatial scales that aquifers span, most of these transformations occur at the vadose zone, a hot spot where electron donors, acceptors, and microbes meet. In the case of dissolved organic carbon, it is rapidly metabolized via aerobic respiration, i.e.,

using dissolved oxygen as the electron acceptor. Similarly, nitrification, the oxidation of ammonium into nitrate, also occurs under aerobic conditions where microbes use ammonia and dissolved oxygen as a means of energy production. Denitrification, i.e., the reduction of nitrate, can also happen in SAT systems, in hotspots where oxygen is depleted, and microbes utilize nitrate to oxidize dissolved organic carbon (Bouwer, 2002; Elkayam et al., 2015).

    Microbial growth and metabolism result in biomass accumulation in the soil pore space, impeding water flow, lowering

infiltration rates, and subsequently hindering aquifer recharge (Bouwer, 2002); process known as *bioclogging*. Operational controls in SAT systems, such as drying and wetting cycles, are imposed to allow air to penetrate the soil, replenish with oxygen, desiccate biomass, and restore infiltration rates (Ben Moshe et al., 2021). More recalcitrant clogging materials can continue accumulating and drying alone might not be enough to recover infiltration rates. At that point, mechanical scraping of the clogging layer is needed to restore the permeability and infiltration rate of the SAT system (Bouwer, 2002; Negev et al.,

2020).

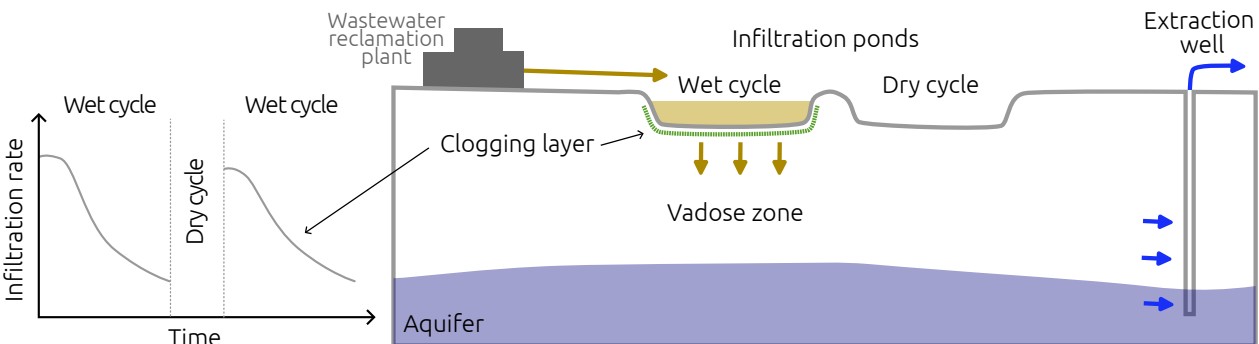

**Figure 1.** SAT drying and wetting cycles schematic. Adopted from Bear and Cheng (2010, Chapter 3.4), Dillon (2005); Grinshpan et al. (2021).

    The downside of the strategy of drying cycles is that it dramatically decreases the total volume of water that can be treated by SAT, as water does not infiltrate the soil during those periods (Bouwer, 2002). Maximizing water input to SAT while maintaining nutrient transformation rates is essential in maintaining SAT operation; however, determining optimal wetting and drying periods relies mainly on the experience of the SAT operators (Sharma and Kennedy, 2017). Efforts to understand the

underlying mechanisms of bioclogging in SAT have combined column experiments (Abel et al., 2014; Ben Moshe et al., 2020),

pore-scale investigations, and numerical models (Srivastava and Jim Yeh, 1992; Soleimani et al., 2009; Berlin et al., 2015; Ben Moshe et al., 2021) that attempt to uncover the dynamics of unsaturated flow, microbial metabolism, and bioclogging.

In this paper, we propose a continuum model to account for microbial metabolism, infiltration, and bioclogging in unsaturated media, distinguishing between active and inactive biomass, with the latter split between labile and recalcitrant fractions (Figure 2). Labile inactive biomass broadly encompasses dead cells, extracellular polymeric substances (EPS), and other microbial products. Recalcitrant inactive biomass refers to biologically inert material that is not readily biodegraded and accumulates within the pore space (Rittmann and MacCarty, 2020). Microbial growth only leads to the formation of the labile fraction, whereas microbial decay leads to the formation of both labile and recalcitrant fractions (Laspidou and Rittmann, 2002; Mannina et al., 2023). While both active and inactive biomass contribute to bioclogging, only active biomass metabolism contributes to nutrient transformations. In biofilms, for instance, most biomass is inactive as more than 90% is EPS (Flemming and Wingender, 2010). Bioengineering models have considered this biomass fractionation (Ni et al., 2011; Rittmann and MacCarty, 2020), but modeling efforts of bioclogging under unsaturated conditions in soils or SAT systems usually do not (e.g., Brovelli et al., 2009; Kildsgaard and Engesgaard, 2001; Clement et al., 1996; Berlin et al., 2015; Mohanadhas and Kumar, 2019; Soleimani et al., 2009; Ben Moshe et al., 2021).

Our model is validated against measurements of the spatial distribution of active and inactive biomass, data collected from unsaturated column experiments by Rosenzweig (2011, Chapter 8). We then use the model to simulate drying cycles to assess their effect on the hydraulic performance of the column and discuss how these results can be used to optimize SAT operation.

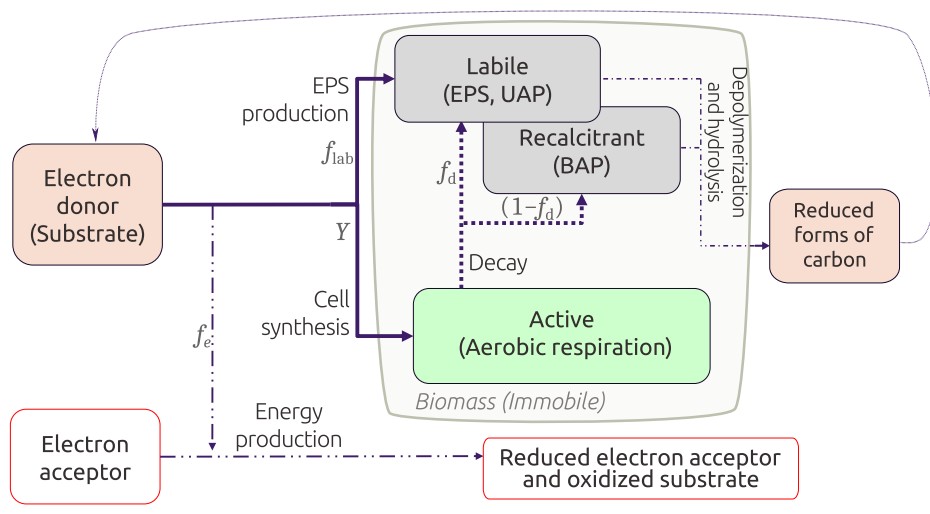

**Figure 2.** Conceptual model of the active and inactive biomass categorization. $f_e$ is the fraction of substrate used for energy production, and $f_s = (1 - f_e)$ is the fraction that goes into biomass generation. Biomass can either be active cells or EPS; thus, $f_s$ is also distributed into the true yield $Y$ and a $f_{lab}$ fraction. Adapted from Ni et al. (2011) and Rittmann and MacCarty (2020).

.

## 2 Methods

### 2.1 Unsaturated flow

The Richards equation is numerically solved to model water flow under unsaturated conditions. We extended this governing equation (Eq. 1) to account for porous media deformations that arises from bioclogging.

$$\left(n\,c(h)\right)\frac{\partial h}{\partial t} + s_w\frac{\partial n}{\partial t} = \nabla\left(\left(\hat{K}_{\mathsf{unsat}}(h)\,\hat{K}_{\mathsf{clog}}(n)\,K_0\right)\nabla\left(h+z\right)\right) \tag{1}$$

In Eq. 1, $n$ represents soil porosity, $s_w$ is the volumetric water saturation, $c(h)$ is the hydraulic capacity function, $K_0$ is the hydraulic conductivity of clean (biomass free) saturated porous media, $h$ is the capillary head, and $z$ is depth. The factor 70 $\hat{K}_{\mathsf{unsat}}(h)$ corresponds to the relative reduction in hydraulic conductivity due to unsaturated conditions, which is calculated using the van Genuchten and Mualem models (van Genuchten, 1980; Mualem, 1976). The factor $\hat{K}_{\mathsf{clog}}(n)$ represents the loss of hydraulic conductivity due to reductions in porosity as a result of bioclogging (Section 2.4).

The van Genuchten model is widely used to describe the water saturation as a function of the capillary head $s_w(h)$. This model is defined in Eqs. 2a and 2b, where $s_{w,\mathsf{sat}}$ is the maximum water saturation, $s_{w,\mathsf{res}}$ is the residual saturation and $s_e$ is an 75 effective saturation function of the capillary head.

$$s_w = s_e(h)\left(s_{w,\mathsf{sat}} - s_{w,\mathsf{res}}\right) + s_{w,\mathsf{res}} \tag{2a}$$
$$s_e(h) = \frac{1}{\left(1 + (-\alpha\,h)^\eta\right)^m} \tag{2b}$$

In Eq. 2b, $\alpha$, $\eta$ and $m = 1 - \frac{1}{\eta}$ are empirical constants that depend on the soil and are determined experimentally. Under unsaturated conditions, $h < 0$, whereas under saturated conditions, the matric head is replaced by pressure head and $s_e = 1$. 80 The Mualem model is defined in Eq. 3 and it is used to calculate the relative hydraulic conductivity of soil under unsaturated conditions $\hat{K}(h)$.

$$\hat{K}(h) = \sqrt{s_e}\left(1 - \left(1 - s_e^{1/m}\right)^m\right)^2 \tag{3}$$

The capillary capacity $c(h)$ is defined as the derivative of the water saturation with respect to the capillary head. This function is calculated analytically from the van Genucthen and Mualem models as shown in Eq. 4.

$$c(h) = \frac{ds_w}{dh} = \alpha\,\eta\,m\left(s_{w,\mathsf{sat}} - s_{w,\mathsf{res}}\right)(-\alpha\,h)^{\eta-1}\left(1 + (-\alpha\,h)^\eta\right)^{-m-1} \tag{4}$$

Although it is expected that unsaturated flow parameters (i.e., the van Genuchten-Mualem model parameters) will change as biomass accumulates, our proposed model does not capture or solve for such changes, hence, keeping the factors $\hat{K}_{\mathsf{clog}}(n)$ and $\hat{K}_{\mathsf{unsat}}(h)$ independent of each other is one of the simplifications we made in the model conception.

These equations are numerically solved using OpenFOAM (Weller et al., 1998) building upon the solver `RichardsFoam`
(Orgogozo et al., 2014; Orgogozo, 2022).

## 2.2 Reactive transport of dissolved constituents

Eq. 5 shows a general expression for the reactive transport of a dissolved constituent $i$ with aqueous concentration $C_i$. $q$ is the specific (Darcy) water discharge, $D$ is the dispersion coefficient, and $\mathcal{R}_i$ is a reaction term that encompasses all the sources and sinks of each component on the reaction network. In this case, where only aerobic respiration is considered, $i$ can be either dissolved organic carbon (DOC, as the electron donor) or dissolved oxygen ($O_2$, as the electron acceptor).

$$\frac{\partial n\, s_w\, C_i}{\partial t} = \nabla \cdot (D\, \nabla C_i) - \nabla \cdot (q\, C_i) + n\, s_w\, \mathcal{R}_i \tag{5}$$

Definitions for $\mathcal{R}_i$ are provided in Eq. 6. The DOC sinks correspond to their consumption by aerobic respirators, whereas sources include hydrolysis and depolymerization of inactive biomass. Aerobic respiration consumes dissolved oxygen, and it is replenished according to the unsaturated conditions that allow air to replenish.

$$\mathcal{R}_{\text{DOC}} = -\, r_{\text{DOC}|\mathbf{ah}} X_{\mathbf{ah}} + k_{\text{hyd}|\mathbf{lab}} X_{\mathbf{lab}} + k_{\text{hyd}|\mathbf{rec}} X_{\mathbf{rec}} \tag{6a}$$

$$\mathcal{R}_{\text{O}_2} = -\, \alpha_1\, r_{\text{DOC}|\mathbf{ah}} X_{\mathbf{ah}} + M n \left( s_a C_{\text{O}_2|\text{sat}} \zeta_{\text{O}_2} \right) \tag{6b}$$

In Eq. 6, $r_{\text{DOC}|\mathbf{ah}}$ is the rate of DOC utilization by aerobic heterotrophs, and $k_{\text{hyd}|\mathbf{lab}}$ and $k_{\text{hyd}|\mathbf{rec}}$ are the rates of labile and recalcitrant biomass hydrolysis, respectively. $X_j$ represents the mass of biomass $j$ per representative elemental volume. The subindex $\mathbf{ah}$ indicates aerobic heterotrophs active biomass, $\mathbf{lab}$ denotes inactive labile biomass, and $\mathbf{rec}$ represents the recalcitrant inactive biomass. $\alpha_1$ is a unit conversion factor between electron donor and acceptor based on the stoichiometry of DOC oxidation. $s_a$ is the volumetric air saturation, $M$ is the oxygen phase transfer rate (i.e, from the air to the water phase), $C_{\text{O}_2|\text{sat}} = 9\text{mg/L}$ is the dissolved oxygen concentration at saturation, and $\zeta_{\text{O}_2} = 1 - C_{\text{O}_2}/C_{\text{O}_2|\text{sat}}$ is a term ensuring that oxygen dissolution stops as saturation is reached.

Details on the rate of nutrient utilization $r$ are described in Eq. 7, where $\hat{q}_{\mathbf{ah}}$ denotes the maximum substrate utilization rate for aerobic heterotrophs, $K_{\text{DOC}|\mathbf{ah}}$ is the half-reaction constant for substrate consumption, and $K_{\text{O}_2|\mathbf{ah}}$ is the half-reaction constant for electron acceptor utilization. $r$ follows dual-Monod kinetics, thus, transformation rates are proportional to microbial content and the concentration of both electron donor and acceptor (Brovelli et al., 2009; Bae and Rittmann, 2000; Rittmann and MacCarty, 2020).

$$r_{\text{DOC},\mathbf{ah}} = \hat{q}_{\mathbf{ah}}\, \frac{C_{\text{DOC}}}{K_{\text{DOC}|\mathbf{ah}} + C_{\text{DOC}}}\, \frac{C_{\text{O}_2}}{K_{\text{O}_2|\mathbf{ah}} + C_{\text{O}_2}} \zeta_X \tag{7}$$

The term $\zeta_X$ is the microbial growth-limiting term by pore-space availability, which is defined in Eq. 8, where $\rho_X$ is the mass density of biomass in aqueous media, $n_0$ the porosity for clean porous media and $n_{\min}$ the minimum possible porosity

after bioclogging. Numerically, $\zeta_X$ ensures that biomass accumulation does not exceed the pore-space that is available in the REV. The term $\rho_X$ represents the biomass density with units of dry mass per wet volume. Under the macroscopic scale of our model, it is interpreted simply as a scaling factor between biomass content and the reduction in hydraulic conductivity (Kildsgaard and Engesgaard, 2001). $\rho_X$ is the most sensitive parameter in bioclogging simulations and it is used as a fitting parameter.

$$\zeta_X = 1 - \frac{X_{\mathbf{ah}} + X_{\mathbf{lab}} + X_{\mathbf{rec}}}{\rho_X \left(n_0 - n_{\min}\right)} \tag{8}$$

### 2.3 Microbial growth and biomass

Microbial biomass is considered immobile, and its evolution over time depends on two terms: a reaction term $\mathcal{R}$ that encompasses their growth and decay, and a diffusion-like term that represents microbial expansion rate towards neighboring volumes (Tronnolone et al., 2018).

$$\frac{\partial X_j}{\partial t} = \nabla \cdot \left(\kappa \nabla X_j\right) + \mathcal{R}_j \tag{9}$$

In Eq. 9, the subindex $j$ refers to either active or inactive biomass. Active biomass refers to one of the metabolic groups that could be modeled; this case will be limited to only aerobic heterotrophs (**ah**). Inactive biomass can be labile (**lab**) or recalcitrant (**rec**). The term $\kappa$ represents the cell diffusivity and its value can range between $10^{-13}$ and $10^{-10}\,\mathrm{m}^2\,\mathrm{s}^{-1}$ (Tronnolone et al., 2018). A representative value of $\kappa = 10^{-11}\,\mathrm{m}^2\,\mathrm{s}^{-1}$ was selected, and we checked that model results were not significantly sensitive to this parameter. The term $\mathcal{R}_j$ encapsulates the reaction terms affecting the biomass fractions; definitions are given below in Eq. 10.

$$\mathcal{R}_{\mathbf{ah}} = \left(Y_{\mathbf{ah}}\, r_{\mathsf{DOC},\mathbf{ah}} - b_{\mathbf{ah}}\right) X_{\mathbf{ah}} \tag{10a}$$

$$\mathcal{R}_{\mathbf{lab}} = f_{\mathbf{lab}}\, r_{\mathsf{DOC},\mathbf{ah}} X_{\mathbf{ah}} + f_d\, b_{\mathbf{ah}} X_{\mathbf{ah}} - k_{\mathrm{hyd}|\mathbf{lab}} X_{\mathbf{lab}} \tag{10b}$$

$$\mathcal{R}_{\mathbf{rec}} = \left(1 - f_d\right) b_{\mathbf{ah}} X_{\mathbf{ah}} - k_{\mathrm{hyd}|\mathbf{rec}} X_{\mathbf{rec}} \tag{10c}$$

In Eq. 10a, $Y$ represents the true yield, which is the fraction of substrate that is converted into active biomass, and $r_{\mathsf{DOC}}$ is the substrate utilization rate (Eq. 7). Microbial decay is modeled as a first-order reaction with a rate of $b$, leading to the generation of both types of inactive biomass. The fraction of active biomass that decays into the labile pool is $f_d$ (Eq. 10b) and the remnant $(1 - f_d)$ decays into the recalcitrant pool (Eq. 10c, Figure 2). Hydrolysis and depolymerization of inactive biomass are expressed as first-order reaction processes with rate $k_{\mathrm{hyd}}$, where $k_{\mathrm{hyd}|\mathbf{rec}} < k_{\mathrm{hyd}|\mathbf{lab}}$. These processes represent DOC sources reflected back in Eq. 6. Labile inactive biomass like EPS is also generated during microbial metabolism, and this fraction is accounted for in the first term on the right hand side of Eq. 10b (Figure 2).

## 2.4 Bioclogging and permeability loss

During growth, biomass fills up the empty spaces in soil that allow water flow through, which results in a decrease in the soil's effective porosity. Therefore, porosity is linked with biomass content following Eq. 11, where $\sum_j X_j$ is the sum of active and inactive biomass.

$$n = n_0 - \frac{1}{\rho_X} \sum_j X_j \tag{11}$$

Several macroscopic models have been proposed to describe permeability changes in terms of changes in the soil's effective porosity under saturated conditions. Hommel et al. (2018) presented a compilation of such porosity-permeability models and concluded that a simple power law is a good first approximation since no further assumptions are required for sub-REV processes. Eq. 12 shows the bioclogging model adopted based on a Kozeny-Carman equation (Hommel et al., 2018; Saavedra Cifuentes et al., 2023), where $\hat{K}(n)$ is the relative hydraulic conductivity penalized by porosity reduction. The lower limit on hydraulic conductivity accounts for the assumption that biomass itself is also permeable (Pintelon et al., 2012; Hassannayebi et al., 2021).

$$\hat{K}(n) = \frac{K_0 - K_{\min}}{K_0} \left( \frac{n - n_{\min}}{n_0 - n_{\min}} \right)^3 + \frac{K_{\min}}{K_0} \tag{12}$$

We have considered that changes in hydraulic conductivity due to unsaturated conditions are independent from changes due to bioclogging. This allowed us to split the hydraulic conductivity $K(h,n)$ into separate terms $K = K_0 \hat{K}(h) \hat{K}(n)$. However, biomass content can change unsaturated flow parameters in soils (Rosenzweig, 2011; Volk et al., 2016).

## 2.5 Experimental data

Column experiments under unsaturated conditions were conducted and reported by Rosenzweig (2011, Chapter 8). The column was $60\,\mathrm{cm}$ long and filled with Caesarea sand. Its saturated hydraulic conductivity $K_0 = 1.24\,\mathrm{cm/min}$ was determined using a constant head permeameter (Dane and Clarke Topp, 2002), and the unsaturated flow parameters in Table 1 were determined using the hanging column method and the pressure plate method (Rosenzweig et al., 2012). The porous medium was sterilized and inoculated with *Pseudomonas Putida F1*, an obligate aerobe (Palleroni, 2015). An initial flow rate of $1\,\mathrm{mL/min}$ was imposed to guarantee unsaturated conditions from the beginning of the experiment. Microbes were fed with a diluted lysogeny broth equivalent to a DOC concentration of 100 mg/L. The column experiment was run for 23 days. Over time, bioclogging drove the infiltration rate lower than the injection rate, and water ponding at the top of the column was evident. Another column packed with sterile sand and fed with distilled water served as a control experiment.

Measurements of matric head using tensiometers were recorded every 30 minutes at six different depths. Similarly, water content from TDR sensors was extracted every 4 hours. Water content was also measured gravimetrically at the end of the experiment. A more detailed description of the column setup is provided in Appendix A1 and the data recorded is shown in

**Table 1.** Column experiment characteristics and unsaturated soil properties.

| Parameter | | Value | |
|---|---|---|---|
| *Geometry* | | | |
| Length | | 0.60 m | |
| Diameter | | 0.08 m | |
| Initial flow rate | | 1.0 mL/min | $(1.67 \times 10^{-8}\,\mathrm{m^3/s})$ |
| *Soil characteristics* | | | |
| Grain size range | | 105 to 590 $\mu$m | $(1.05 \text{ to } 5.90 \times 10^{-4}\mathrm{m})$ |
| Saturated hydraulic conductivity | $K_0$ | 1.24 cm/min | $(2.07 \times 10^{-4}\ \mathrm{m/s})$ |
| *Unsaturated flow parameters* | | | |
| Residual water saturation | $s_{w,\mathsf{res}}$ | 0.0312 | |
| Maximum water saturation | $s_{w,\mathsf{sat}}$ | 1.0 | |
| van Genuchten fitting exponent | $\eta$ | 7.26 | |
| van Genuchten fitting parameter | $\alpha$ | $2.79\ \mathrm{m^{-1}}$ | |

Experiments from Rosenzweig (2011, Chapter 8).

Appendix A2. At the end of the experiment, the columns were dismantled and the top sections were destructively sampled. Profiles of microbial counts, protein content, and water content were measured over depth. Microbial counts and protein content measurements served as a proxy for active and total biomass, respectively. Microbial counts were not directly translated into the units of simulated active biomass, so only their trends are compared. In contrast, protein content measurements are translated to total inactive biomass, considering the fractionation of proteins and carbohydrates in biomass. In Eq. 13, $X_{\mathbf{in}|\text{experiment}}$ represents the total inactive biomass measured at the end of the column experiment, Protein content is the measured data and Protein percentage $= 0.68$ is based on the data from Malamis and Andreadakis (2009).

$$X_{\mathbf{in}|\text{experiment}} = \frac{\text{Protein content}}{\text{Protein fraction}} \tag{13}$$

## 2.6 Computational domain and model parameterization

Reactive transport equations were coupled and solved simultaneously with unsaturated flow using the same finite-volume mesh. The computational domain corresponded to a 1D representation of the column experiment, discretized with element size $\Delta z = 0.01\,\mathrm{m}$. The time step was variable and bounded to $\Delta t \leq 10\,\mathrm{s}$ to ensure the Courant–Friedrichs–Lewy (CFL) condition remained less than 1.0 during all simulations. We compiled data for microbial growth parameters from previous models in the literature and adopted those biogeochemical parameters for our model. Values and sources are provided in Table 2.

**Table 2.** Microbial growth, metabolism, and biomass parameters. Values compiled from various sources: Berlin et al. (2015); Taylor and Jaffé (1990b); Mostafa and Van Geel (2012); Thullner et al. (2004); Brovelli et al. (2009); Kildsgaard and Engesgaard (2001); Rittmann and MacCarty (2020).

| Parameter | | Value | |
|---|---|---|---|
| *Aerobic respirators* | | | |
| Max. specific rate of substrate utilization | $\hat{q}_{\mathbf{ah}}$ | $7.0\,\mathrm{d}^{-1}$ | $(8.1 \times 10^{-5}\,\mathrm{s}^{-1})$ |
| Microbial decay (die-off) rate | $b_{\mathbf{ah}}$ | $0.30\,\mathrm{d}^{-1}$ | $(3.5 \times 10^{-6}\,\mathrm{s}^{-1})$ |
| True yield | $Y_{\mathbf{ah}}$ | $0.49$ | — |
| Half-reaction constant for electron donor | $K_{\mathrm{DOC}|\mathbf{ah}}$ | $10.0 \times 10^{-3}$ | $\mathrm{kg/m}^3$ |
| Half-reaction constant for electron acceptor | $K_{\mathrm{O}_2|\mathbf{ah}}$ | $3.5 \times 0.5^{-3}$ | $\mathrm{kg/m}^3$ |
| *Inactive biomass* | | | |
| Labile fraction of dead biomass | $f_d$ | $0.80$ | — |
| Fraction of substrate used to form EPS | $f_{\mathbf{lab}}$ | $0.18$ | — |
| Hydrolysis rate of EPS and labile biomass | $k_{\mathrm{hyd}|\mathbf{lab}}$ | $0.17\,\mathrm{d}^{-1}$ | $(1.97 \times 10^{-6}\,\mathrm{s}^{-1})$ |
| Hydrolysis rate of recalcitrant biomass | $k_{\mathrm{hyd}|\mathbf{rec}}$ | $0.017\,\mathrm{d}^{-1}$ | $(1.97 \times 10^{-7}\,\mathrm{s}^{-1})$ |
| Biomass density | $\rho_x$ | $10$ | $\mathrm{g/L}$ |

Initial conditions for active biomass reflected a small homogeneous inoculum along the domain, setting $X_{\mathbf{ah}}(z, t = 0) = 0.1\,\mathrm{mg/L}$. Inactive biomass is set to zero at the beginning of each simulation. Since the hydraulic conductivity field is tied to the total biomass distribution, it begins at its clean-state value and decreases as the simulation advances, reflecting biomass generation. This is a crucial difference with the SAT optimization model proposed by Ben Moshe et al. (2021), where a low hydraulic conductivity layer is imposed in the uppermost part of the column. In our model, as in other biomass substrate formulations (e.g., Taylor and Jaffé, 1990b), the formation of such a low-permeability layer is the result of the biogeochemical-unsaturated flow coupling, and it is not an imposed constraint on the model.

The boundary conditions at the top of the column are modified during wet and dry cycles. During wetting periods, a head gradient is initially imposed on the top boundary to obtain a target flow of 1mL/min. Since the hydraulic conductivity $K$ decreases over time due to clogging, the head gradient required to maintain the target flux increases over time, thus, we recalculate it at every time step. The topmost layer in the system is clogged if the head gradient needed to achieve the flux target results in a positive head value at the boundary. In that case, we switch the top boundary condition to a constant head ($h = 0$) to represent ponding. Drying periods are represented imposing a zero-gradient at the top boundary. A constant unit gradient is set for $h$ to ensure free flow at the bottom boundary of the column. The computational domain and a summary of boundary conditions are shown in Figure 3.

To calibrate our model, we fitted the simulated biomass distribution to the experiment measurements of inactive biomass profiles. We found that $\rho_X = 10\,\mathrm{g/L}$ gave a good fit and agreed in order of magnitude with values from the literature (Thullner et al., 2004; Kildsgaard and Engesgaard, 2001; Mostafa and Van Geel, 2012; Clement et al., 1996; Caruso et al., 2017).

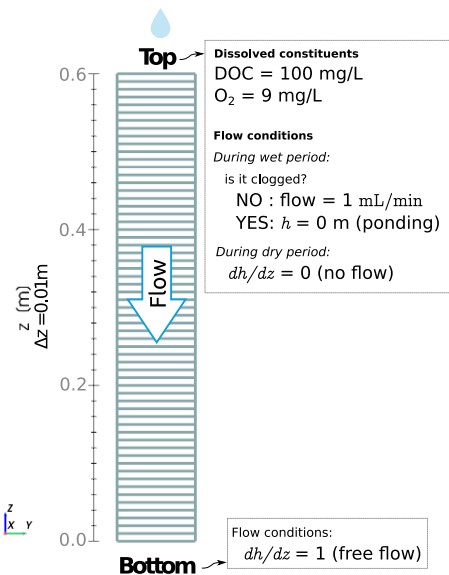

**Figure 3.** Computational domain for simulating column experiments. The top boundary condition flips between fixed flow and constant head depending on whether the upmost layer is clogged and water is ponding. The bottom boundary represents a free-draining condition.

## 2.7 Drying periods and hydraulic efficiency

We investigated the impact of different wetting periods and dry/wet time ratios using the same model. We ran experiments using $t_{\mathrm{wet}}$ between 150 to 1450 minutes and under dry/wet time ratios ranging from 0.25 to 6. The total infiltrated water volume per unit area obtained from these simulations is reported in Appendix A4 (Figure A12). For further exploration, we fixed the wetting time to $450\mathrm{min}$ and tested dry/wet time ratios ranging from 6 to 0, where the latter case corresponded to a constant water injection. We also explored the effect of different drying times, maintaining a set dry/wet time ratio of 4.5. These simulations were run up to 80 days until biomass fractionation reached a steady state. Finally, the **long-term hydraulic loading rate** is used to calculate hydraulic efficiency, defined as the total infiltrated volume per unit area divided by the duration of the experiment (Bouwer, 2002).

# 3 Results

## 3.1 Comparison with experimental depth profiles

Figure 4 displays the biomass distributions over depth at the end of the column experiment. Panel a shows the active biomass profile calculated from the numerical model, superimposed on the experimental data. The experimental data and the model prediction align regarding their spatial distribution, as aerobic heterotrophs accumulate near the top of the column. This trend is expected because the maximum dissolved oxygen and organic carbon concentration are present at this boundary (Taylor and Jaffé, 1990a). Microbial metabolism leaves fewer electron donor and acceptor available to sustain microbes in deeper sections, resulting in fewer active cells in the deeper parts of the column. Nutrient limitation arises from two interlinked factors: microbial metabolism depletes dissolved constituents, reducing concentrations in lower regions, while bioclogging and reduced inflow into the column hinder nutrient supply. The accumulation of inactive biomass near the column's surface, depicted in Panel b, reflects this trend, as it is a byproduct of heterotrophic metabolism. Samples from the column feeding solution and effluents were taken only at the end of the experiment for total organic carbon analysis and that analysis showed 89% removal of the substrate. In the simulations, 99% of the input DOC is consumed. Although there was no continuous monitoring of effluent substrate during the experiment, this end-point measurement provides a reasonable check, considering that SAT systems generally lead to near-complete DOC consumption. Outflow DOC concentration over time is plotted in Figure A16.

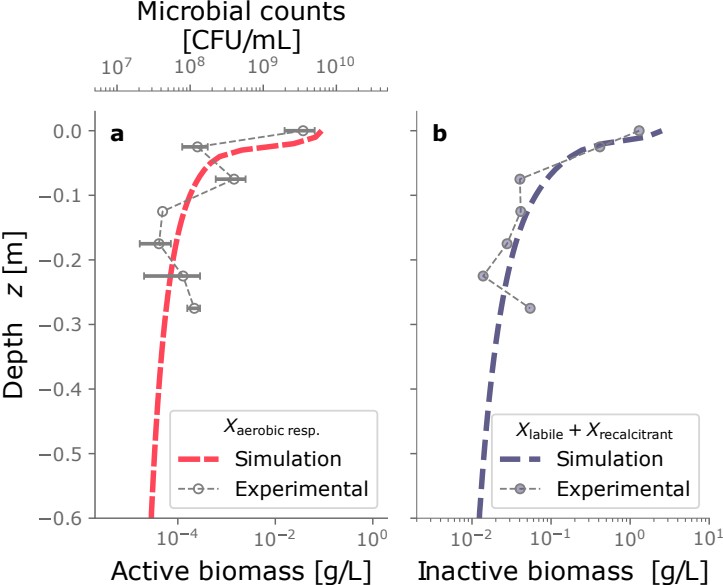

**Figure 4.** Biomass spatial distribution from experimental data and the numerical model. Panel **a** shows the active biomass over depth. Panel **b** displays the inactive biomass, which consists of the sum of labile and recalcitrant fractions.

## 3.2 Evolution of biomass over time

From the numerical model, we tracked the evolution over time of the water infiltration rate and the total biomass in the domain (Figure 5a - b). Early times in the system correspond to the exponential stage of microbial growth, characterized by high substrate transformation rates facilitated by the unclogged porous media. This is reflected in the fast increment of total active biomass that peaks at $2.1 \times 10^{-2} \mathrm{kg/m^2}$ at $t = 4.0\,\mathrm{d}$. Simultaneously, inactive biomass accumulates, and bioclogging becomes noticeable as the decline of water influx into the system (Figure 5a). As biomass clogs the system, less water infiltrates the column, and less substrate is available to sustain the already generated active biomass. Its decline ensues, as evidenced in the tail of the plot after $t = 4\,\mathrm{d}$. At this point, inactive biomass becomes the dominant fraction of the total biomass. Inactive biomass continues to accumulate and sustains the clogged state. After $t = 20\,\mathrm{d}$, active biomass content has fallen to a relatively constant value, and the system reaches a steady state where inactive biomass dominates. By the end of the experiment ($t = 23\,\mathrm{d}$), influx water dropped to 3% of the original 1 mL/min injection imposed. Active, labile, and recalcitrant biomass percentages are 3%, 80%, and 17%, respectively. Even though the column experiment was finalized after 23 days, we continued the simulation to check that the steady state was conserved.

## 3.3 Drying cycles

We investigated the impact of cycling drying and wetting periods using the same model conditions. We simulated wetting cycles by introducing water into the top of a column and letting it infiltrate, followed by drying cycles without any water flux. The amount of water influx used for wetting cycles was consistent with the experimental conditions. The simulation was extended to 80 days to observe multiple alternations between wetting and drying cycles until the system reached an apparent steady state. Figures 5c and d show the evolution over time of water influx and total biomass under dry/wet cycles of 1800/450 min, respectively. This is just one of the time combinations tested and further explained in Section 3.4. Water and nutrient input are interrupted during the drying periods, reflected as the dips in influx in Figure 5c. This triggers periods of active biomass decay evident in the repetitive declines in Figure 5d. The fast exponential growth phase that characterized the constant wetting experiment (Figure 5b) is strongly attenuated under the cycling of drying periods. In this case, total active biomass peaks at $9.7 \times 10^{-3} \mathrm{kg/m^2}$, half of the maximum achieved under constant wetting conditions. Moreover, the time to reach the peak in total active biomass is delayed to $t = 11\,\mathrm{d}$. This behavior is expected because substrate delivery is interrupted, and active cells are limited to consuming hydrolyzed inactive biomass nearby. Just as microbial growth is restricted, bioclogging also occurs more slowly, as apparent from the rate at which the water influx decreases over time. Still, inactive biomass builds up in the domain, and a clogged stationary state is reached.

This particular comparison between constantly wet and alternating dry/wet cycles of 1800-450 min was introduced because both simulations achieve a long-term hydraulic loading rate of $2.3 \times 10^{-2} \mathrm{m/s}$. However, the extent of the drying periods and the dry/wet time ratio affect the hydraulic performance. In fact, we are interested in evaluating how imposing those drying periods in the system translates to an operational benefit in increasing the long-term hydraulic loading rate.

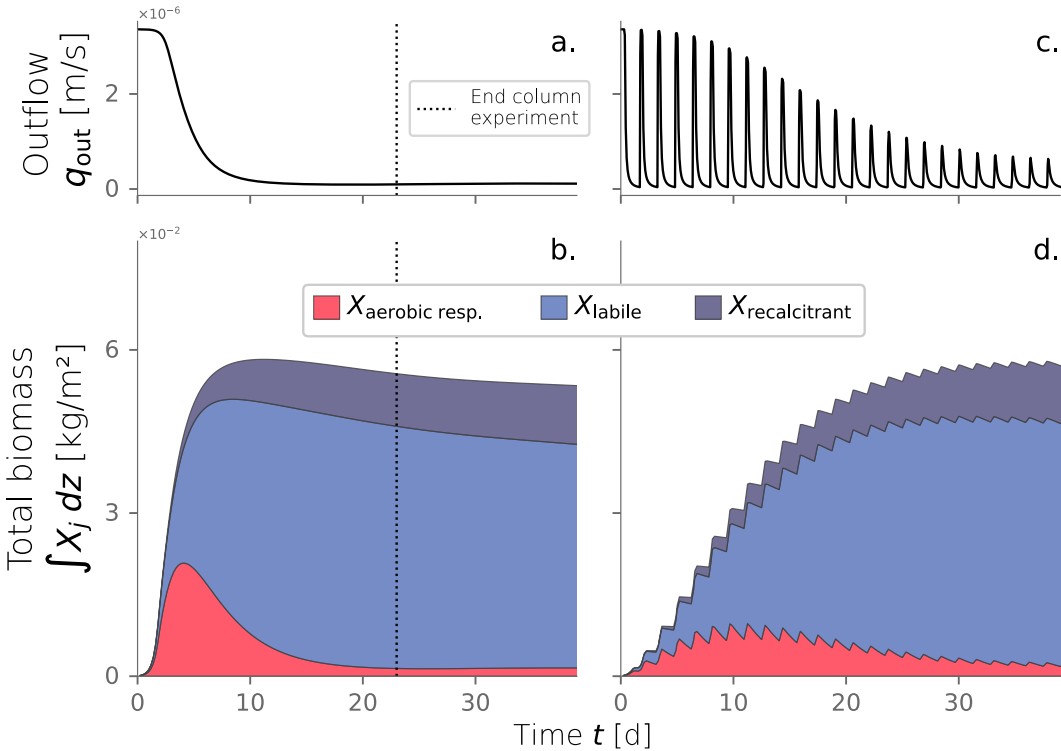

**Figure 5.** Water outflux (a) and fractionation of the total biomass over time (b) for a column experiment simulation under constantly wet conditions. Water outflux (c) and total biomass over time (d) under dry/wet cycles of 1800-450 min.

## 3.4 Link between dry/wet time ratios and hydraulic performance

In order to explore the effect of dry/wet time rations in the hydraulic response of an SAT, we ran simulations under a constant wetting time $t_{wet}$ of 450 minutes and tested a range of drying time. From each dry/wet time ratio, we calculated the total volume of water infiltrated over time and the long-term hydraulic loading rate. The water volume infiltrated over time was calculated by integrating the water flux over time (Figure 6a), and it serves as a hydraulic performance metric. Unsurprisingly, a constantly wet condition performs better than any of the dry/wet cycling strategies for early times ($t < 60d$). This is explained by the

fact that the initial condition of the soil is a clean, unclogged state, so imposing drying periods reduces the amount of water infiltrating the system. The benefits of drying periods become evident only after a long-term operation when bioclogging has intensified and can be offset. After $t < 60d$, the cumulative volumes of infiltrated water associated with some of the simulations surpass that of the constantly wet case (Figure 6a). In fact, a time exists at which a dry/wet cycling strategy pays off because the infiltrated volume of water surpasses the constantly wet alternative, and, therefore, it can be considered more advantageous

from a hydraulic optimization perspective. However, not every dry/wet time ratio has the potential to surpass the performance of the always-wet simulation eventually. Very short drying periods display low infiltration volumes even after long-term operation,

and the rate at which this infiltrated water volume grows over time (i.e., the infiltration rate at the end time) remains equal to or lower than that of a fully bioclogged system.

Another way to summarize hydraulic efficiency comes from the long-term hydraulic loading rate in Figure 6 b. This metric is defined in Section 2.7 as the ratio between the total water volume infiltrated by the end of the simulation over the simulated time (in this case, $80\,\text{d}$) (Bouwer, 2002). This time was chosen because it corresponds to the point when an apparent steady state was observed for all simulations. The constant wet simulation's hydraulic loading rate is set as a reference to determine which dry/wet ratios achieve better hydraulic performance. In our particular set-up, dry/wet ratios higher than 4.5 outperform the always-wet strategy after $t = 80\,\text{d}$.

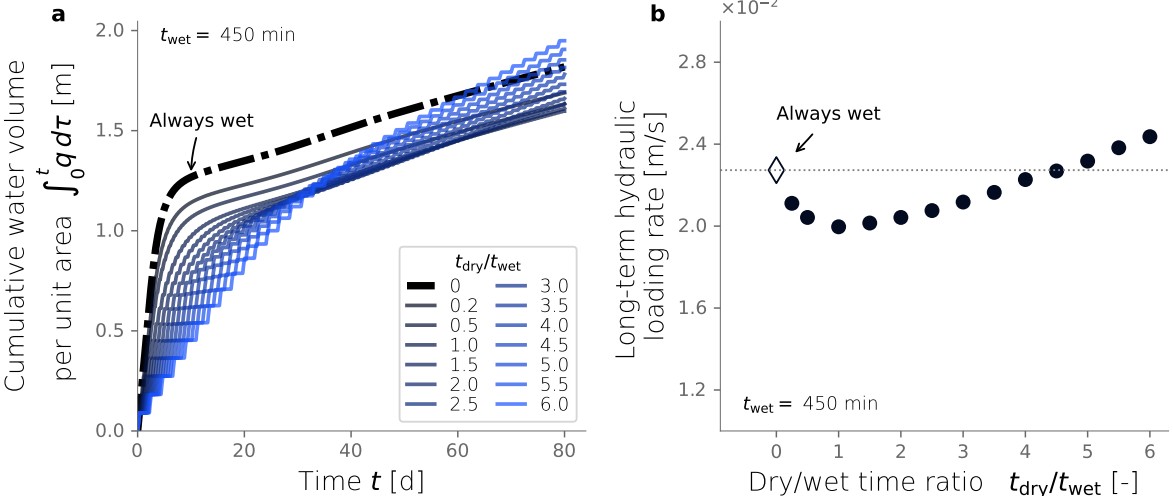

**Figure 6.** Cumulative water volume infiltrated over time (a). Long-term hydraulic loading as a function of the dry/wet time ratio (b). All simulations were run under wet time periods of 450min

Interestingly, longer drying periods also promote higher total biomass contents, a trend evident in Figure 7 a. We observed that dry/wet time ratios greater than 2.0 exhibit greater biomass accumulation than constantly wet simulation, which is again used as a comparison reference. In fact, there is a statistically significant correlation between dry-wet time ratio and total biomass in the domain ($p-$value $< 0.001$). A clue to this link is found in the spatial distribution of the active biomass at the end of the simulations, illustrated in Figure 7 b. These results indicate that drying periods facilitate the growth of higher amounts of active biomass in the deeper regions of the column. In contrast, the characteristic distribution of biomass that develops from the constantly wet condition, with sharp accumulation near the top of the column, becomes more uniform under dry/wet conditions. Interestingly, the depth profiles of inactive biomass (Figures 7c-d) remain moderately unmodified relative to the constantly wet case. The most noticeable distinction is that higher contents of inactive biomass accumulate in deeper regions, and lower contents remain close to the top boundary. This follows the same trend observed from the distribution of active biomass.

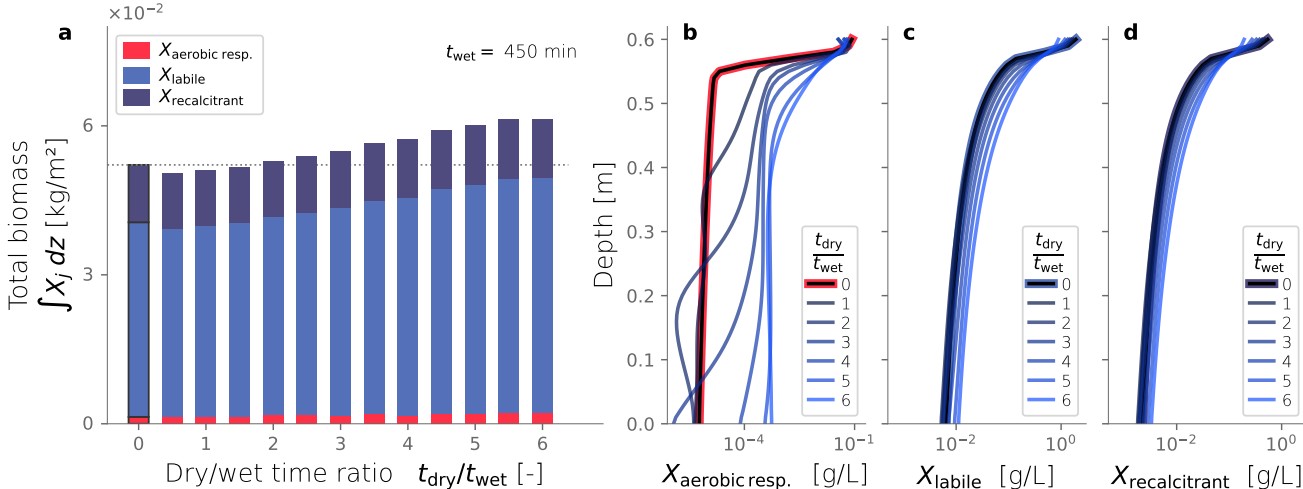

**Figure 7.** Biomass fractionation for multiple dry-wet time ratios (**a**). Active (**b**), inactive labile (**c**) and inactive recalcitrant (**d**) biomass profiles over depth at steady-state for multiple dry-wet time ratios.

## 3.5 Link between dry time duration and hydraulic performance

Up to this point, we evaluated the effect of dry/wet time ratios while keeping a constant wetting time of $450\,\mathrm{min}$. Hereafter, we consider a constant dry/wet time ratio of $4.5$, and we evaluate the impact of different drying times over the same simulation setup. These simulations are also ran until $t_{end} = 80\mathrm{d}$, time at which an apparent steady-state was reached. Figure 8 shows the cumulative water volume (a) and the long-term hydraulic loading rate (a) of simulations under different dry time durations. These two metrics reflect hydraulic performance, and most simulations outperform the constantly wet case used as a comparison reference. For example, the cumulative water infiltrated in cases with $t_{dry} > 2000\mathrm{min}$ is greater than that of the constantly wet case. In fact, around $t \approx 30\mathrm{d}$, the simulations with the longest dry times surpass the volume of water infiltrated achieved under a constantly wet condition. This same behavior is reflected in the long-term hydraulic loading rate of our simulations, thus, in this sense, most simulations hydraulically outperform the constantly wet case.

In Figure 8b, a trend is observed where longer drying periods lead to higher long-term hydraulic loading rates. This is because longer dry times can restore the infiltration rates to a state closer to the clean, unclogged condition. However, this benefit only goes so far, as drying and oxidation of recalcitrant inert biomass can take significantly more extended periods, making it impractical for SAT purposes. Longer drying times achieve higher long-term hydraulic loading rates, but that benefit plateaus and must reach a maximum. In theory, there exists a long enough dry time where all the labile and recalcitrant biomass has desiccated and oxidized, restoring the perfectly unclogged state. However, even longer drying periods will result in low long-term infiltration rates. In the extreme case of $t_{dry} \to \infty$, i.e., the trivial constantly dry case, the long-term loading rate must return to zero. In any case, these results show that hydraulic optimization is not limited to finding an optimal dry/wet time ratio but also involves finding the drying time itself.

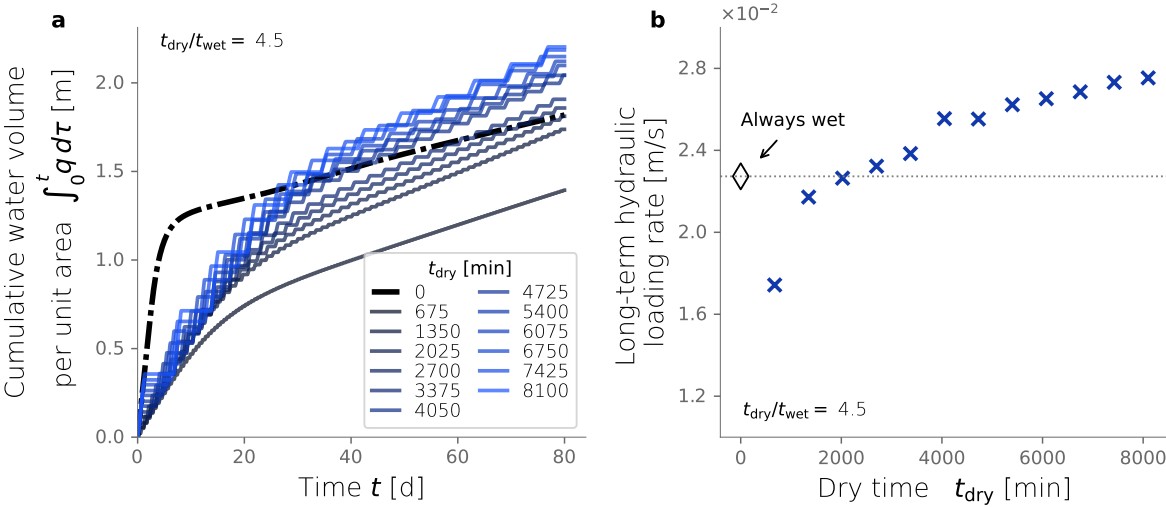

**Figure 8.** Long term hydraulic loading as a function of the drying time, under constant dry-wet time ratio of 4.5

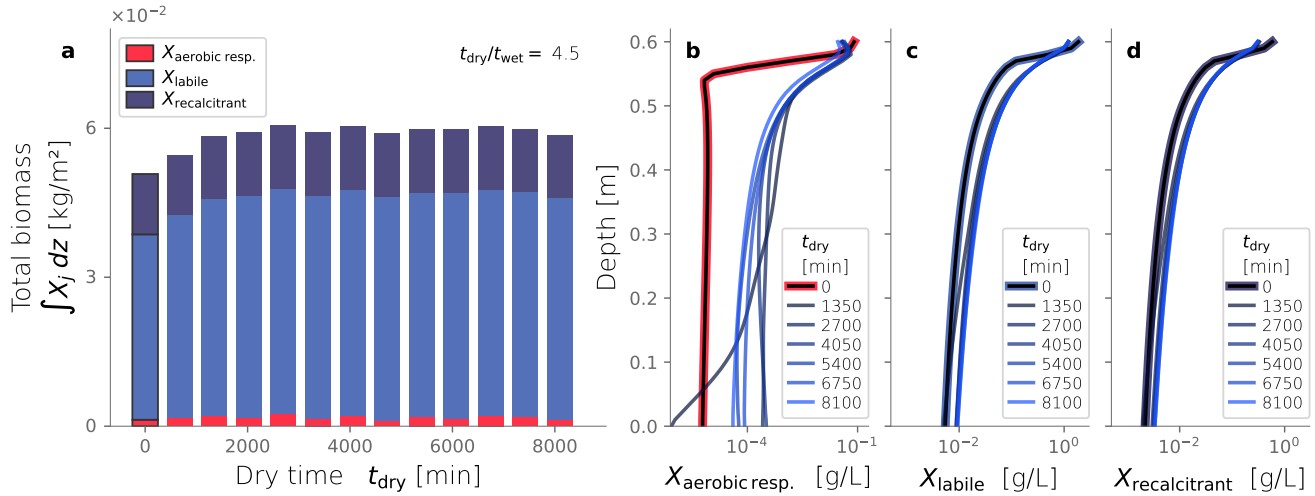

**Figure 9.** Biomass fractionation (a) and profiles over depth at steady-state (b-d) varying the dry time while keeping the dry-wet time ratios equal to 4.5.

Finally, Figure 9a shows the total biomass and its fractionation at the end of each simulation, varying the drying time. All cases with drying cycles exhibit a total biomass accumulation higher than the constantly wet reference case ($t_{dry} = 0$). However, longer drying times do not seem to have a discernible effect on the total biomass accumulation or its fractionation ($p-$value $> 0.01$). This result starkly contrasts the correlation found between total biomass and dry/wet time presented in the previous section. The profiles over depth of each biomass fraction are presented in Figures 9b-d. All cases appear to converge

to the same spatial distribution of active biomass, reflecting the similar profiles of labile and recalcitrant inactive biomass.

Therefore, neither the spatial distribution nor the total amount of active biomass is sensitive to the drying time duration, but they are to the dry/wet time ratio.

## 4    Discussion

Alternation of drying cycles is a common and inexpensive strategy to revert bioclogging in infiltration ponds; however, determining optimal operation strategies based on a physically-based understanding of the underlying processes remains elusive. In-situ and experimental data on microbial communities' spatial distribution, growth, and dynamics under unsaturated conditions are scarce, while mathematical models oversimplify the tight coupling between flow and biogeochemistry. SAT optimization depends on the specific goals set for the wastewater treatment and reuse system. For instance, more significance can be given to carbon removal or nitrate mineralization, while water storage and hydraulic performance are prioritized in other cases. Our exploration focuses on the latter, evaluating the drying times and the dry/wet time ratios that maximize the long-term hydraulic loading rate. We demonstrate the existence of a dry/wet time ratio threshold above which this strategy outperforms a constant influx of water. Our simulations suggest that the dry/wet ratio also controls the spatial distribution of active biomass in the soil column, as higher ratios promote a more homogeneous distribution over depth. This is in contrast to the typical bioclogging profile with heterotrophic biomass accumulation in the topmost regions. The link between biomass distribution and dry/wet time ratio is explained by the possibility of nutrient-rich water penetrating deeper regions while the growth of active biomass at the topmost layer is constrained.

The length of wet and dry cycles in SAT systems is adjusted based on the current system conditions and constraints. In practice, operators adapt the cycle lengths based on experience and limited real-time system monitoring (Sharma and Kennedy, 2017). The model we present provides a basis for improving operations via quantitative simulation and prediction and it could be applied adaptively with real-time or near-real-time ingestion of operating data. However, we do not expect that the optimal drying and wetting times that we find for the column experiments coincide with field SAT applications. In fact, wet and dry periods are longer in full scale SAT systems and, for the Shafdan SAT in particular, wet periods are typically 1-2 days and dry periods are 2-4 days (Sharma and Kennedy, 2017; Idelovitch and Michail, 1984). Scaling our model up from experiment to field will require revisiting the parameterization made for the column experiment. Namely, biochemical parameters (e.g., substrate consumption and die-off rates, yield, etc.) will likely need to be adjusted because the organic carbon source in that case is more recalcitrant than the growth medium used to feed the column.

We chose the length of the simulations as the characteristic time scale needed for the definition of long-term hydraulic loading, as it accounts for both the initial high flows into the clean column and the later low flows after clogging occurs. For SAT applications, this time scale should be chosen to span system resets, meaning it should cover the dry periods when there is no water inflow and until the system infiltration capacity is restored either by a much longer drying time or some mechanical intervention.

The conceptual model proposed for soil biomass fractionation is a simple representation that fills a gap in bioclogging modeling by distinguishing between active and inert biomass. This differentiation is crucial because only active cells contribute

to substrate utilization, while inert by-products contribute to soil bioclogging but not to nutrient transformations. Mostafa and Van Geel (2012) highlighted this gap and recommended a post-processing step to calculate bioclogging. They suggested that inactive biomass accounted for twice the active biomass content and that a fifth of that inactive portion corresponded to the recalcitrant fraction. Active and inert biomass fractions vary over time, as the active fraction dominates early growth stages and the inert fraction accumulates over time and dominates at steady states. Brangarí et al. (2018) proposed a model that tracks the development of different biomass compartments for EPS, active, and dormant cells, along with extracellular enzymes. While this biomass representation is a valuable resource for research and testing of hypotheses, the downside of having complex descriptions of biomass is that it becomes difficult to link them with field data and practical applications (Brangarí et al., 2018). Striking a balance between capturing essential phenomena and ensuring that the model is simple enough to be validated with available data can be challenging. Our proposed approach to compartmentalize active and inactive biomass is a commonly applied strategy in bioreactor modeling applications (Ni et al., 2011). Ultimately, this description for biomass fractionation can be used to describe other metabolic pathways, e.g., nitrification and denitrification, so they can extend the numerical model capabilities.

We focused our parameter exploration on the calibration of the biomass density term $\rho_X$ because previous research had shown the sensitivity of bioclogging models to this parameter (Brovelli et al., 2009, Figures A7 and A8). The value we found for $\rho_X$ was similar to other values in the literature. The composition of bulk biomass in the simulation changes over time because at very early times, the bulk biomass is dominated by the active fraction, but inactive biomass accumulates in the porous medium and becomes the dominant biomass fraction later. Therefore, the accumulation of inactive biomass means that lighter-weight EPS become a larger fraction of the total biomass over time, and the cell density within the biomass correspondingly decreases over time.

We conducted additional simulations to investigate the sensitivity of other less studied parameters, such as the spreading rate $\kappa$ (Figures A9 and A10) and the biomass permeability $K_{min}$ (Figure A11), which affect the biomass spatial distribution. However, changes to these parameters did not significantly alter the resulting biomass profiles. Considering that the column experiment dealt with a single bacterial species growing within a fairly homogeneous media, we retrieved values from the literature for other biochemical parameters, such as the substrate consumption rate and the half-saturation constant. These values fitted the experimental biomass distribution profiles well, giving a fair representation of the heterotrophic activity under unsaturated conditions in our simulations. In addition, EPS and other inactive biomass are expected to maintain a hydrated micro-environment for active cells (Or et al., 2007), which might explain why a dramatic change in metabolism parameters was not required. This might not be valid for more abrupt water saturation changes, as experimental data suggests that microbes adapt to water stress periods, devoting more resources to EPS generation to retain moisture and increase survival under very dry conditions (Flemming and Wingender, 2010; Roberson and Firestone, 1992). We considered the proportion of substrate used for cell synthesis ($Y$) and for EPS generation ($f_{lab}$) to be constant over time and independent of the water saturation as the wet periods in our experiments still yielded unsaturated conditions; however, the variability of these growth parameters from saturation to desiccation needs further exploration.

Wetting periods in infiltration ponds correspond to flooding conditions, as opposed to the wetting period in our column experiment, which corresponded to a slow injection aimed at keeping unsaturated conditions during all times in the experiment.

Similarly, the DOC concentration of a water reclamation plant effluent is much lower (around 15-20 mg/L) (Idelovitch et al., 2003) than the DOC concentration from the lysogeny broth used to feed the *Pseudomonas Putida* in the column. In this column experiment, the DOC source was a growth broth characterized with a concentration of 100 mg/L in the simulations. We tested the influence of setting other influent concentrations and found that the system reached the same end-state of accumulation of biomass near the top clogging the system. Nevertheless, a more in depth exploration of particular settings that characterize

SAT systems effectively is a matter of modifying the boundary conditions of the model that we have presented. In particular, running a column experiment with the actual effluent of a wastewater treatment plant would be immensely informative because the DOC bio-availability is much lower than that of the growth broth. Relative to the column experiments, lower yields are expected in SAT systems primarily due to the nature of the dissolved organic carbon (DOC) typically present in secondary-treated wastewater effluent. This DOC is generally more refractory and less readily metabolized by microorganisms compared

to the growth substrate used in the controlled column experiments. This difference in substrate bio-availability would not necessarily lead to less clogging in SAT systems in the long term, but rather to slower clogging initially. The accumulation of refractory fraction of inactive biomass that we found to be a key driver of bioclogging will still occur: accumulation of recalcitrant material will eventually dominate the clogging process, leading to the clogged end state we observe and simulate in our model.

Available experimental data in this field are generally limited to monitoring infiltration rates over time under always-wet conditions. Biomass profiles over depth are typically available only at a single time point at the end of experiments, as obtaining them requires destroying the sand column. This limitation partially constrains the testing of our conceptual model. However, we use the available experimental data to validate our numerical simulations initially and then explore hypotheses related to drying cycles. Our simulations help narrow down the range of optimal conditions, thereby guiding future experimental efforts.

Nevertheless, reevaluation of the model will be necessary as new experimental data become available. A recommended path for future experimentation is to set up conditions analogous to field operation, so combined with our model, we can draw more robust conclusions of hydraulically optimal dry/wet time ratios and drying times apt to SAT system conditions. Likewise, other field conditions not captured by our simulations could be readily integrated into our modeling framework. For example, sunlight drives photosynthesis in the infiltration ponds, leading to dissolved oxygen over-saturation, and the effluent infiltrated

during the day is more oxidized than that during night (Goren et al., 2014). Similarly, evaporation could affect the rate at which biomass desiccates during dry cycles, potentially influencing how quickly the system recovers to a near-clean state and how clogging is mitigated. Another example is the inclusion of fine particle deposition that can be an additional source of clogging of the soil interface. These factors might affect optimal day-to-day operation; thus, all of these variables integrated into the model have the potential to convert this piece of research software into a SAT operation decision-making tool.

Biomass accumulation also affects the water retention of soils (Philippot et al., 2023; Volk et al., 2016; Costa et al., 2018; Colica et al., 2014). For example, experiments from Rosenzweig et al. (2012) showed that EPS content, using xanthan gum as a surrogate, enhanced Caesarea sand's water retention. Two parameters were clearly affected, namely, the maximum water

saturation $s_{w,\text{sat}}$ and the exponent $\eta$ in the van Genuchten model (Eq. 2b). The former was linked to increased soil porosity due to the expansion of xanthan after hydration, whereas the later was linked to the soil's water retention capacity. As data is lacking, we did not explicitly introduce a constitutive relation between biomass content and unsaturated flow properties. Still, this behavior can potentially shift the optimal drying periods to longer than predicted because water retention increases with biomass content.

## 5 Conclusions

Our study explored the alternation of drying cycles in infiltration ponds to mitigate bioclogging and enhance the hydraulic performance of SAT systems. We propose a simple conceptual model for fractionating soil biomass into active and inert compartments, which provides a good fit for experimental data and can be potentially extended to describe other metabolic pathways. We focused our parameter exploration on the biomass density term, which was found to be sensitive and critical for bioclogging modeling. Our simulations showed that the dry/wet time ratio is pivotal in controlling the spatial distribution of aerobic respirators in the soil column. A threshold exists above which this strategy outperforms a constant influx of water. While our findings cannot be directly generalized to field SAT operations due to the dissimilarities between laboratory and real-world conditions, they provide a solid basis for further research and optimization of SAT systems.

*Code and data availability.* Source code for the numerical model is available in Zenodo at Saavedra Cifuentes (2024).

## Appendix A: Appendix

### A1 Column experiment details

Column experiments were carried out and reported in Rosenzweig (2011). The setup consists of an inoculated column and a control. *Pseudomonas Putida F1* was used as the model bacteria because their ability to produce large amounts of biofilms. The control column only contained sterilized sand. Each columns was 60 cm long, built out of twelve 5 cm long 9 cm diameter Plexiglas sections. SCOTCH layers were used to retain sand inside the columns. Drippers at the columns' inlet were equipped with 63 evenly distributed 0.8 mm diameter needles. The feeding solution of the inoculated column consisted of diluted LB broth and distilled water was used for the control. The top six sections of each column were equipped with ports for tensiometers and TDR probes. A schematic of the experimental setup and detail of the sensors installation is shown in Fig. A1. Data collected from the tensiometers is shown in Fig. A2 and water content derived from the TDR probes are shown in Fig. A3.

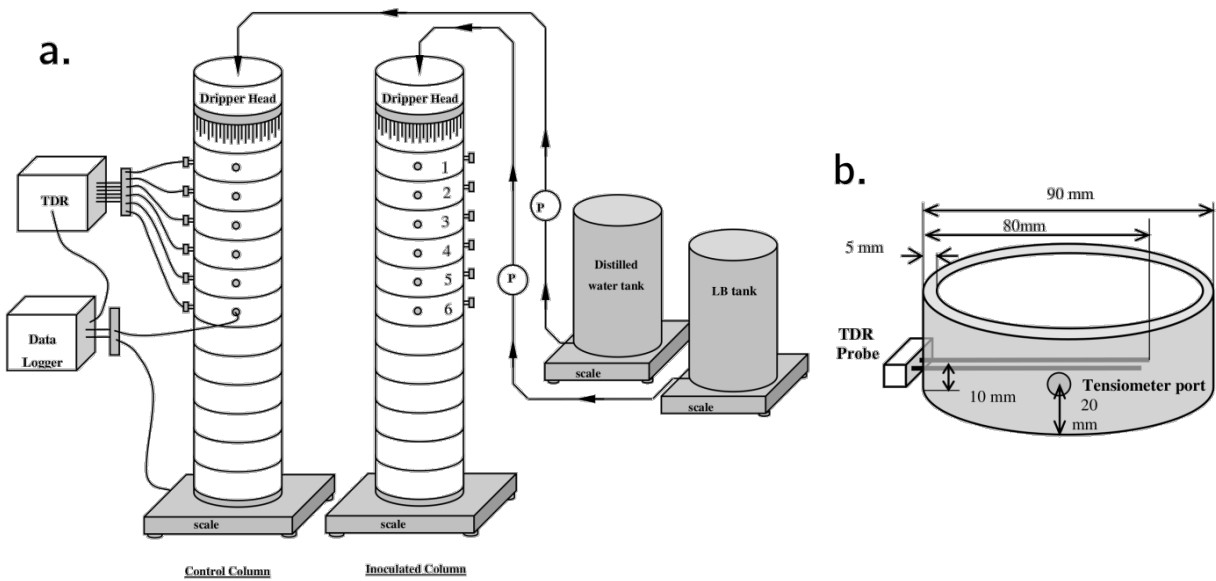

**Figure A1. a.** Schematic of the experimental setup. Control and inoculated columns were fed with distilled water and diluted LB broth, respectively. **b.** Detail of one of the column sections equipped with water content sensors (TDR probes) and tensiometers.

At the end of the experiments, the columns were dismantled and the top six sections were destructively sampled. At the top section, the sand was divided into surface layer (top 0.5 cm) and the rest of the section. Sand at each of the sections was thoroughly mixed before sand samples were collected. Biofilm distribution in the different column sections was characterized by determining the viable bacteria concentration and cells' protein content. Water content at the end of the experiment was determined gravimetrically. Comparison between that measurement, water content from TDR probes and simulation results is show in Fig, A13.

## A2 Experimental data

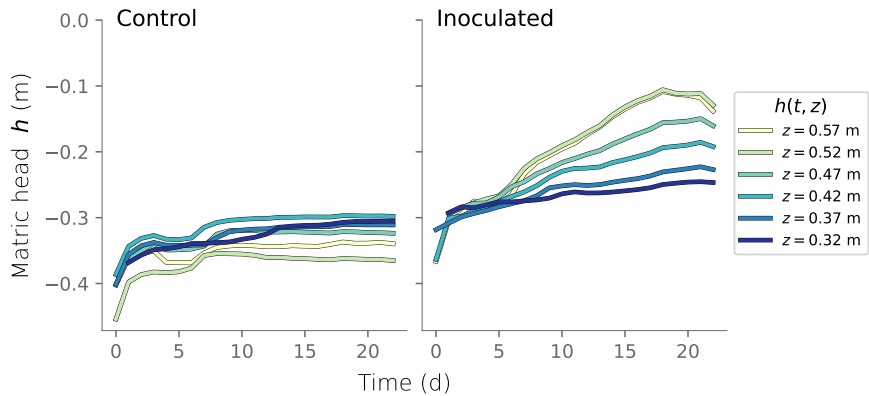

**Figure A2.** Time series of matric head measured at six different depths in the column experiments.

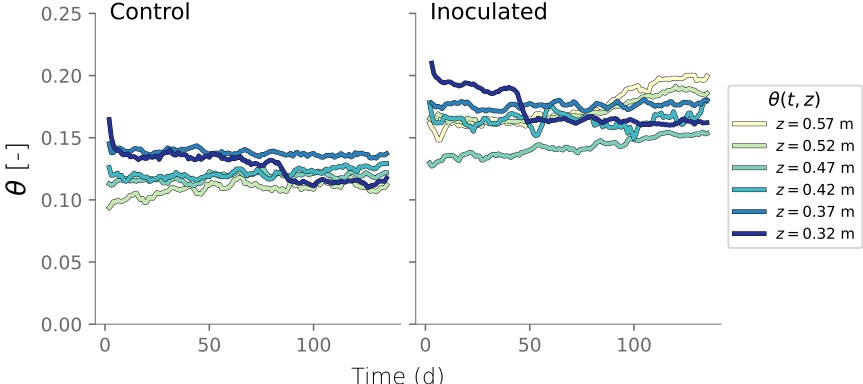

**Figure A3.** Time series of water content measured at six different depths in the column experiments.

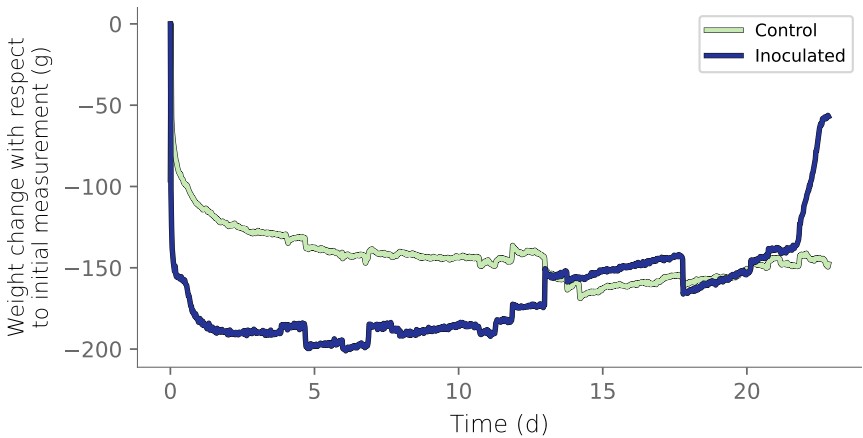

**Figure A4.** Column weight over time relative to the initial reading. Initial weight reduction is attributed to the initial draining of the wet packing of the column.

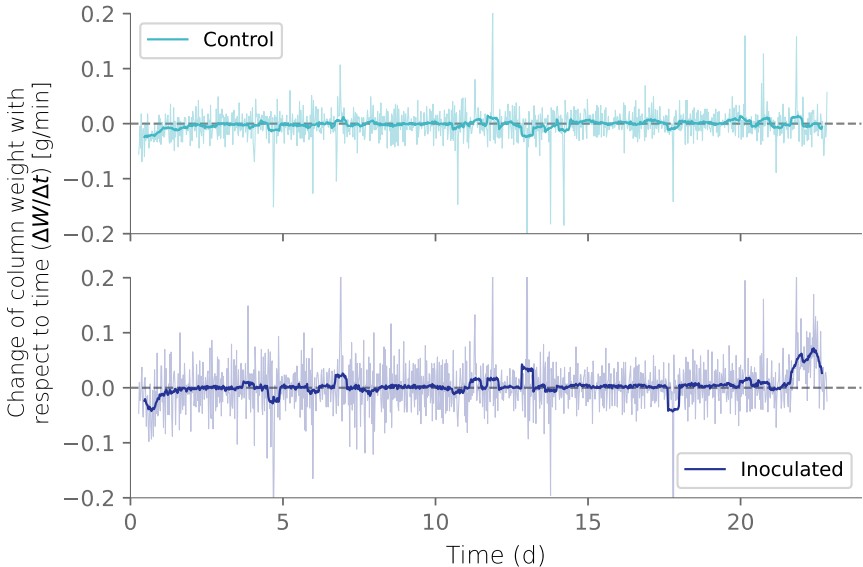

**Figure A5.** Net flux from the columns calculated from the weight differentials over time. Negative net fluxes indicate more water leaving than entering the columns.

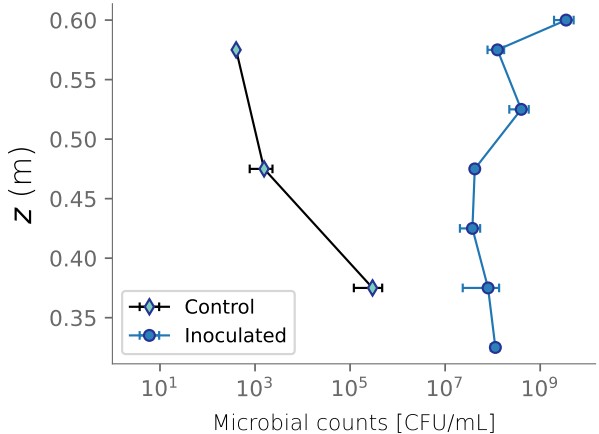

**Figure A6.** Colony formation units (CFU) measured at the end of the column experiment.

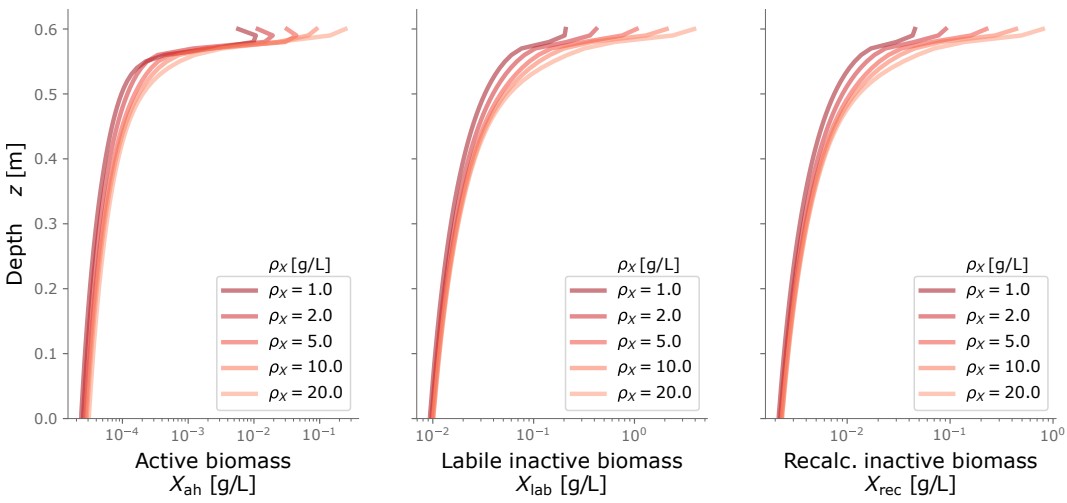

**Figure A7.** Biomass profiles under different values of biomass density $\rho_X$. Typical values are around $10$ g/L.

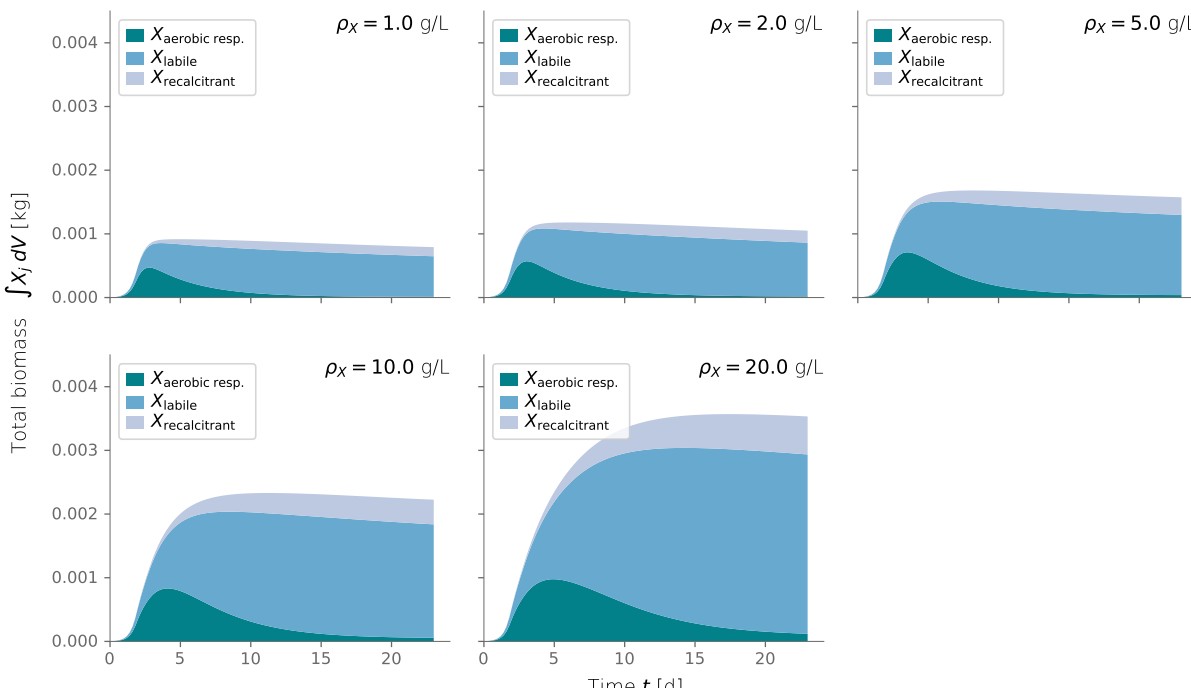

**Figure A8.** Total integrated biomass under different values of biomass density $\rho_X$. Typical values are around $10$ g/L.

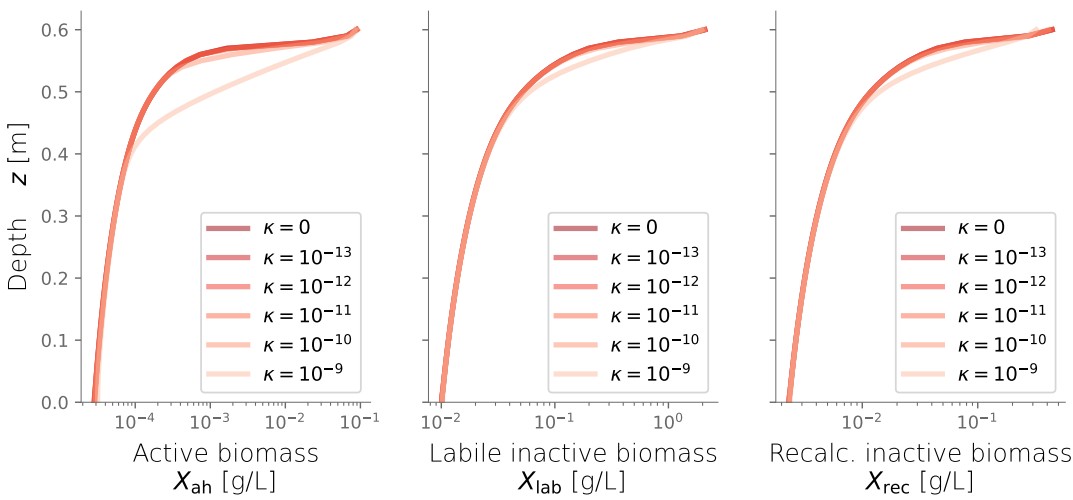

**Figure A9.** Biomass profiles under different values of diffusive growth coefficients $\kappa$. Values over $10^{-10}$ m/s are not expected.

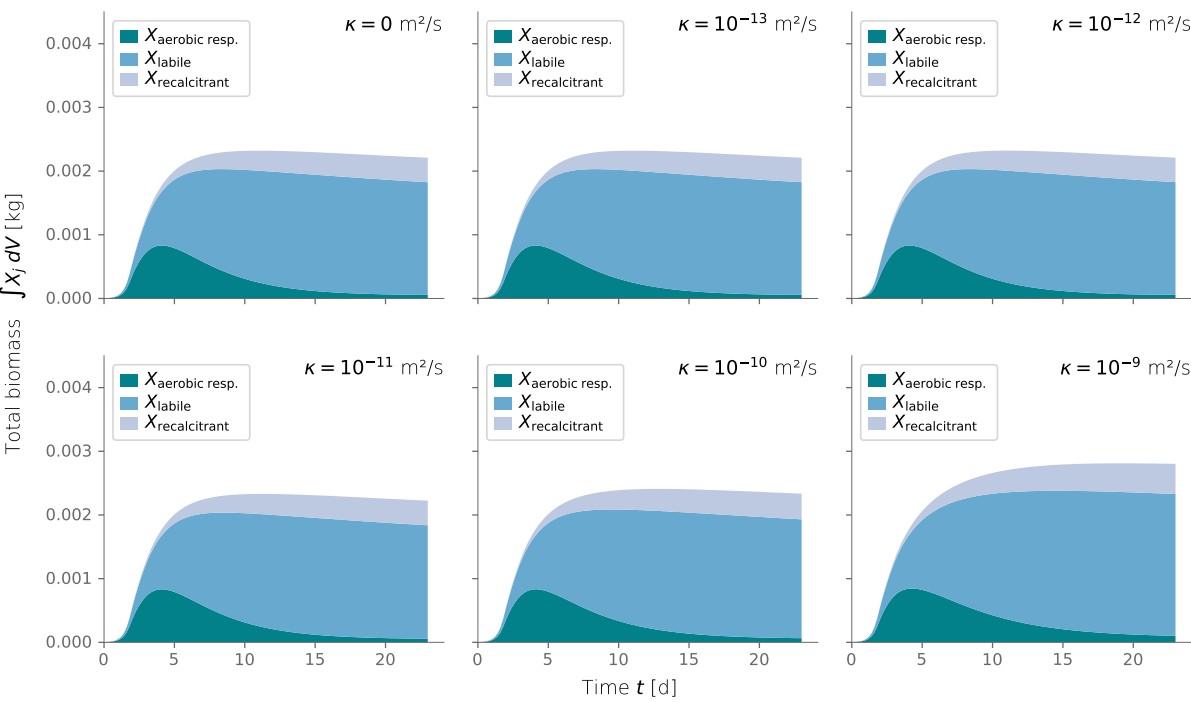

**Figure A10.** Total integrated biomass under different values of diffusive growth coefficients $\kappa$. Values over $10^{-10}$ m/s are not expected.

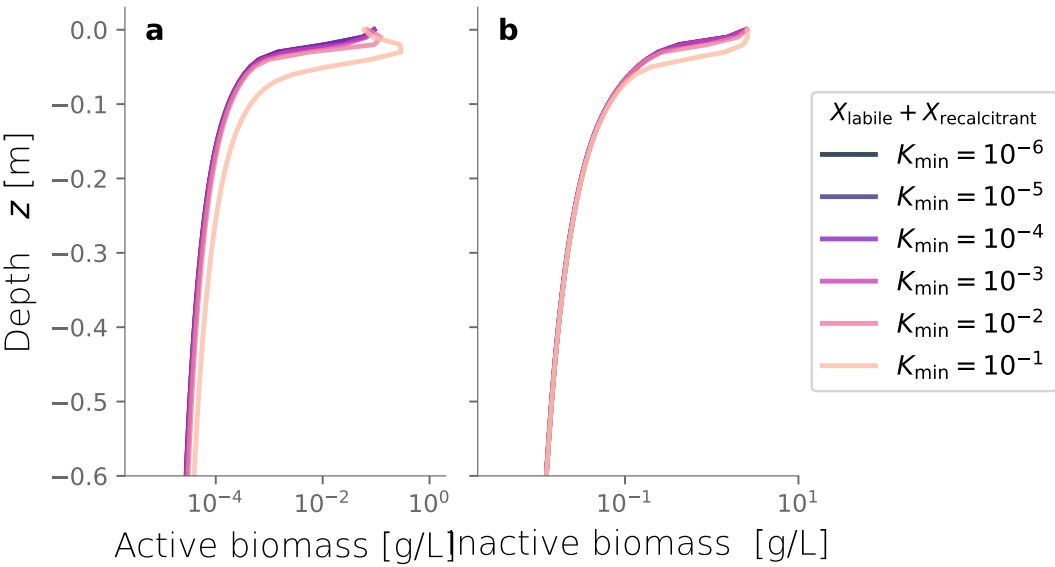

**Figure A11.** Biomass profiles under different values of biomass hydraulic conductivity $K_{min}$. Values are relative to the porous medium's clean saturated hydraulic conductivity.

## A4 Numerical simulations

| $t_{dry}/t_{wet}$ | $t_{wet}$ [min] | | | | | | | | | | | | | |
|---|---|---|---|---|---|---|---|---|---|---|---|---|---|---|
| | 150 | 250 | 350 | 450 | 550 | 650 | 750 | 850 | 950 | 1050 | 1150 | 1250 | 1350 | 1450 |
| 0.00 | 1.81 (Always wet) | | | | | | | | | | | | | |
| 0.25 | | 1.60 | 1.59 | 1.70 | 1.68 | 1.67 | 1.67 | 1.72 | 1.71 | 1.71 | 1.74 | 1.70 | 1.73 | 1.72 |
| 0.50 | 1.43 | 1.54 | 1.60 | 1.64 | 1.67 | 1.61 | 1.64 | 1.67 | 1.68 | 1.69 | 1.70 | 1.68 | 1.70 | 1.72 |
| 0.75 | 1.57 | 1.51 | 1.50 | 1.60 | 1.67 | 1.58 | 1.64 | 1.68 | 1.66 | 1.66 | 1.71 | 1.71 | 1.74 | 1.74 |
| 1.00 | 1.43 | 1.50 | 1.55 | 1.58 | 1.61 | 1.63 | 1.64 | 1.65 | 1.68 | 1.70 | 1.72 | 1.73 | 1.75 | 1.76 |
| 1.25 | 1.31 | 1.50 | 1.47 | 1.56 | 1.63 | 1.60 | 1.66 | 1.70 | 1.69 | 1.74 | 1.77 | 1.72 | 1.75 | 1.79 |
| 1.50 | 1.48 | 1.50 | 1.53 | 1.56 | 1.66 | 1.59 | 1.69 | 1.70 | 1.72 | 1.73 | 1.79 | 1.75 | 1.80 | 1.81 |
| 1.75 | 1.36 | 1.51 | 1.47 | 1.56 | 1.61 | 1.66 | 1.71 | 1.76 | 1.74 | 1.77 | 1.81 | 1.80 | 1.81 | 1.84 |
| 2.00 | 1.26 | 1.53 | 1.54 | 1.56 | 1.65 | 1.67 | 1.68 | 1.76 | 1.77 | 1.77 | 1.83 | 1.83 | 1.84 | 1.88 |
| 2.25 | 1.44 | 1.54 | 1.48 | 1.66 | 1.70 | 1.67 | 1.72 | 1.81 | 1.79 | 1.81 | 1.84 | 1.87 | 1.89 | 1.91 |
| 2.50 | 1.34 | 1.56 | 1.56 | 1.66 | 1.75 | 1.67 | 1.74 | 1.81 | 1.81 | 1.87 | 1.90 | 1.87 | 1.91 | 1.93 |
| 2.75 | 1.51 | 1.58 | 1.49 | 1.67 | 1.70 | 1.75 | 1.77 | 1.87 | 1.84 | 1.86 | 1.92 | 1.89 | 1.94 | 1.97 |
| 3.00 | 1.42 | 1.60 | 1.57 | 1.67 | 1.76 | 1.75 | 1.81 | 1.85 | 1.86 | 1.90 | 1.93 | 1.92 | 1.97 | 1.98 |
| 3.25 | 1.32 | 1.62 | 1.64 | 1.66 | 1.80 | 1.71 | 1.84 | 1.84 | 1.87 | 1.93 | 1.95 | 1.96 | 1.97 | 2.01 |
| 3.50 | 1.50 | 1.64 | 1.58 | 1.67 | 1.75 | 1.82 | 1.87 | 1.91 | 1.88 | 1.94 | 1.98 | 2.00 | 2.00 | 2.04 |
| 3.75 | 1.40 | 1.66 | 1.66 | 1.68 | 1.80 | 1.82 | 1.90 | 1.90 | 1.91 | 1.97 | 2.03 | 2.00 | 2.01 | 2.07 |
| 4.00 | 1.31 | 1.68 | 1.60 | 1.69 | 1.84 | 1.81 | 1.85 | 1.96 | 1.95 | 1.95 | 2.03 | 2.03 | 2.04 | 2.09 |
| 4.25 | 1.49 | 1.70 | 1.68 | 1.80 | 1.88 | 1.80 | 1.90 | 1.95 | 1.96 | 2.00 | 2.06 | 2.06 | 2.12 | 2.15 |
| 4.50 | 1.39 | 1.72 | 1.62 | 1.81 | 1.85 | 1.90 | 1.91 | 2.02 | 1.97 | 2.04 | 2.10 | 2.09 | 2.12 | 2.18 |
| 4.75 | 1.59 | 1.75 | 1.71 | 1.82 | 1.86 | 1.88 | 1.95 | 2.01 | 2.01 | 2.05 | 2.11 | 2.12 | 2.16 | 2.19 |
| 5.00 | 1.49 | 1.76 | 1.64 | 1.82 | 1.94 | 1.88 | 1.99 | 2.08 | 2.05 | 2.12 | 2.17 | 2.15 | 2.22 | 2.27 |
| 5.25 | 1.38 | 1.79 | 1.73 | 1.82 | 1.91 | 1.98 | 2.03 | 2.07 | 2.07 | 2.16 | 2.18 | 2.16 | 2.20 | 2.27 |
| 5.50 | 1.59 | 1.81 | 1.67 | 1.84 | 1.97 | 1.98 | 2.06 | 2.13 | 2.08 | 2.14 | 2.23 | 2.26 | 2.30 | 2.35 |
| 5.75 | 1.48 | 1.85 | 1.75 | 1.84 | 1.98 | 1.99 | 2.10 | 2.15 | 2.09 | 2.24 | 2.22 | 2.26 | 2.29 | 2.34 |
| 6.00 | 1.38 | 1.66 | 1.69 | 1.85 | 1.95 | 1.99 | 2.05 | 2.16 | 2.16 | 2.20 | 2.29 | 2.30 | 2.35 | 2.38 |

**Figure A12.** Total infiltrated water volume per unit area (m) after 80 days simulations. When tdry/twet is zero, the case is always wet. Short tdry/wet ratios result in lower infiltrated volumes, until certain threshold is overcome and better hydraulic performance is achieved.

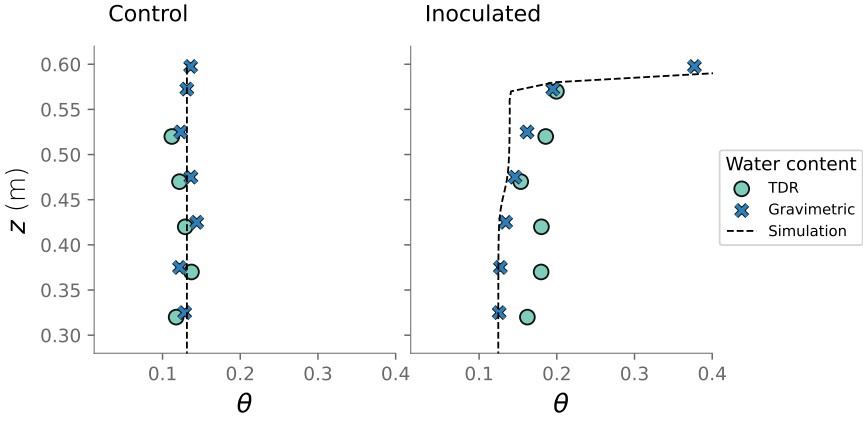

**Figure A13.** Comparison between the water content measured in the column experiment and the water content from the simulation at the end state.

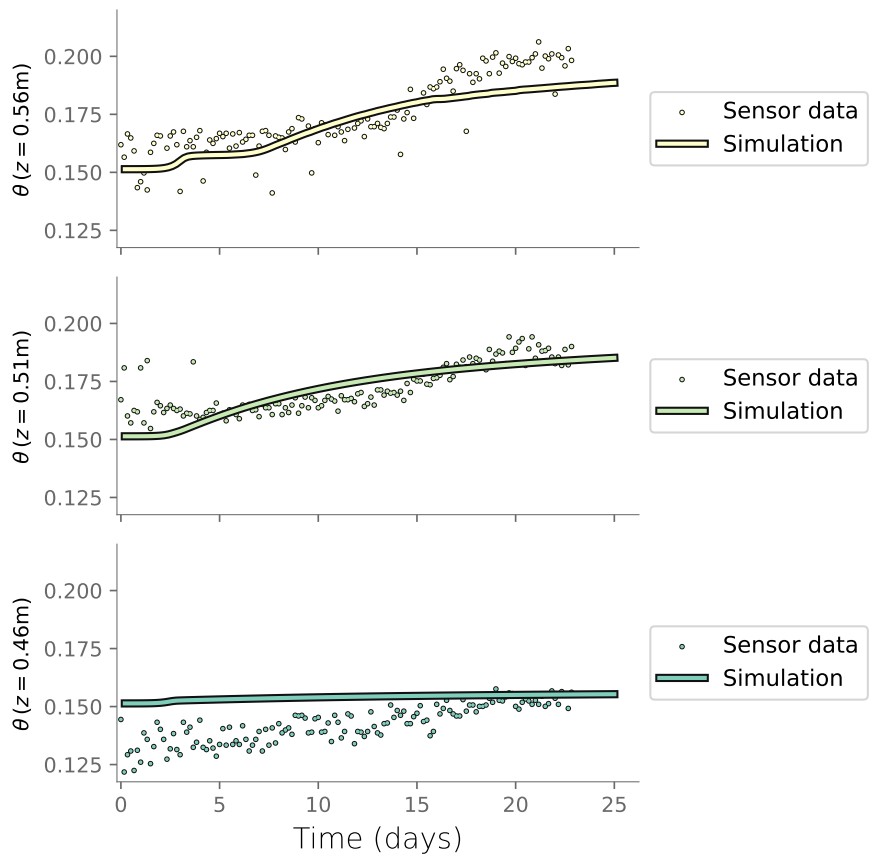

**Figure A14.** Comparison between the water content measured in the column experiment at the three topmost layers and the water content from the simulation.

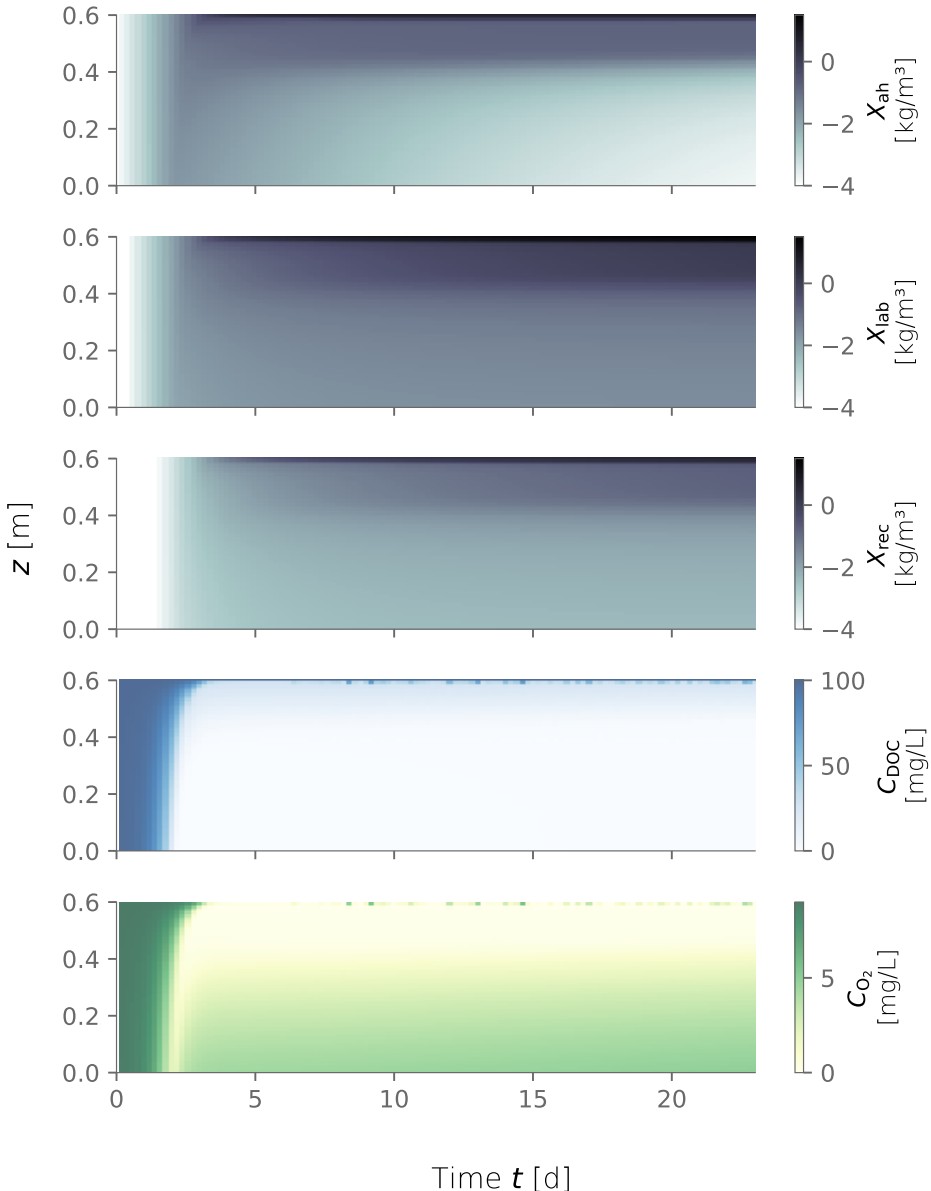

**Figure A15.** Biomass fractions, dissolved organic carbon (DOC) and dissolved oxygen (O₂) over depth and over time for the constantly wet simulation.

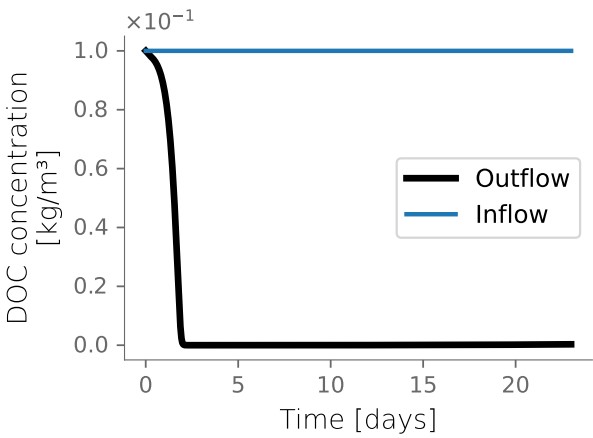

**Figure A16.** Simulated DOC concentration over time for the always wet case.

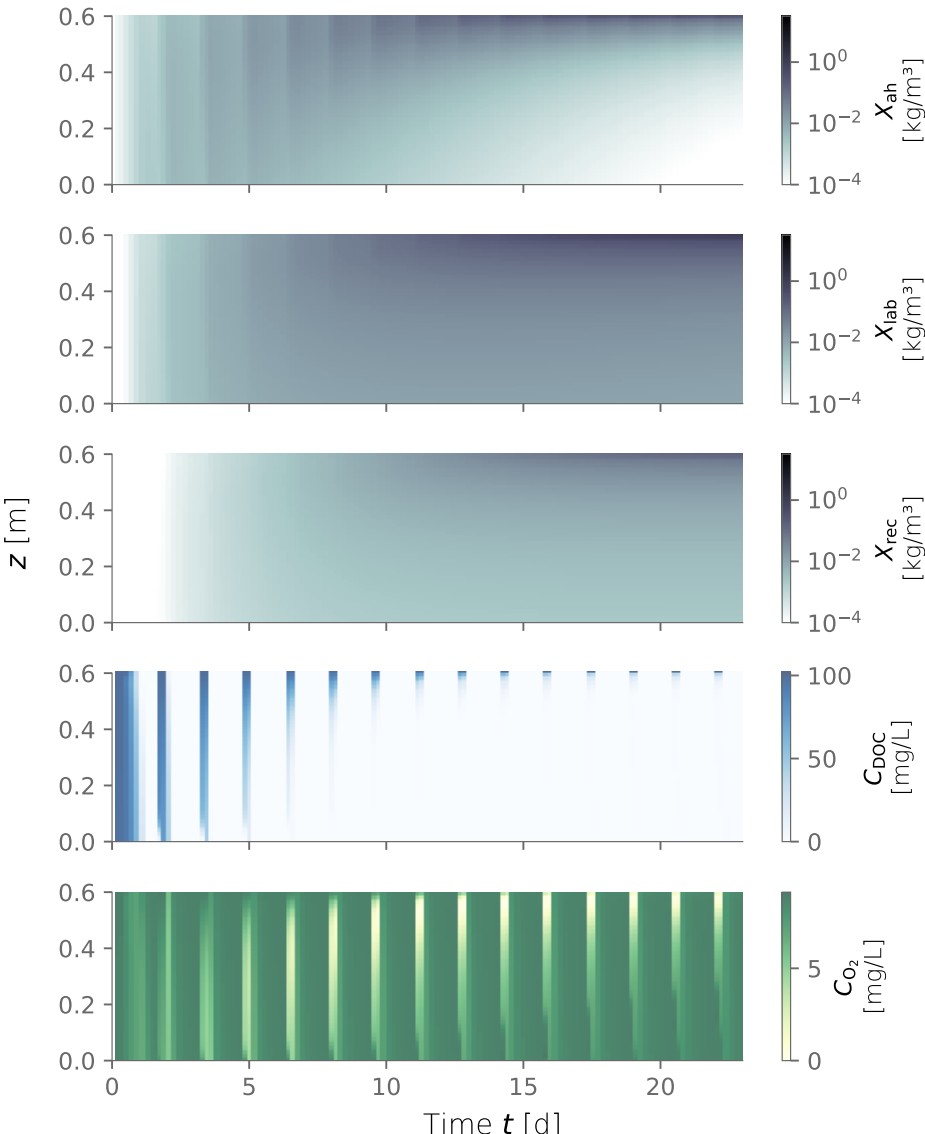

**Figure A17.** Biomass fractions, dissolved organic carbon (DOC) and dissolved oxygen (O₂) over depth and over time for dry/wet cycles of 1800-450 min.

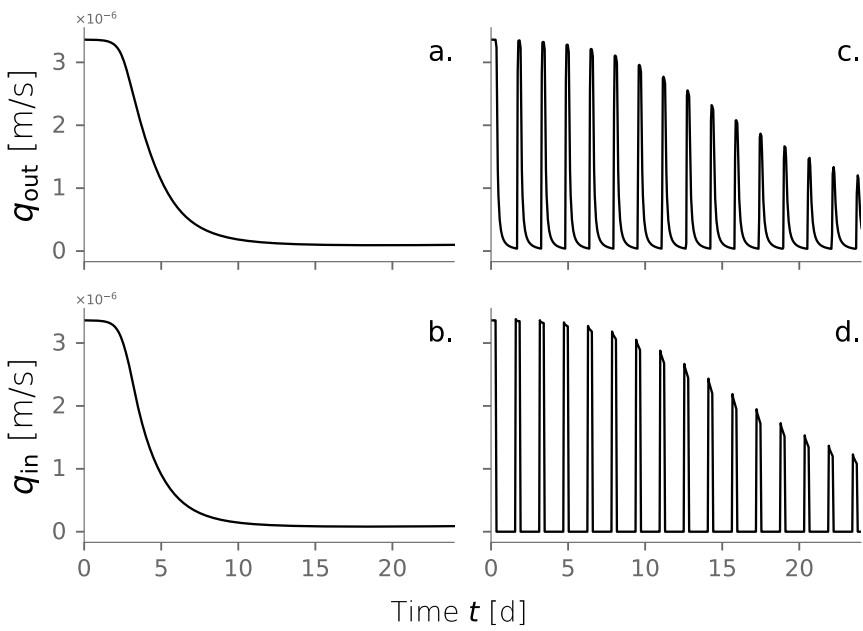

**Figure A18.** Water influx and outflux of the sand column simulations. Figures a-b correspond to the always wet experiment. Figures c-d show dry/wet cycles of 1800-450 min. This is a complement to data shown in Figure 5a and 5c.

*Author contributions.* ESC, AF and AIP planned the research, ESC wrote and ran the numerical model, RR and AF performed the column experiments, ESC, AIP, RR and AF analyzed the data, ESC and AIP wrote the manuscript draft, and ESC, AIP, RR and AF reviewed and edited the manuscript.

*Competing interests.* The authors have no competing interests to declare.

*Acknowledgements.* This work was supported by the Israel-U.S. Collaborative Water-Energy Research Center (CoWERC) via the Binational Industrial Research and Development Foundation (BIRD) Energy Center, grant EC-15.

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
