# Peer review of "Continuum modeling of bioclogging of soil aquifer treatment systems segregating active and inactive biomass"

_Hydrology and Earth System Sciences, 2024_

## Author Comment (AC1)

**Response to Comments – Referee #1**

*I think that the paper is of good general quality, with clear and well-structured writing, and a valuable scientific contribution overall. After having read (and quite enjoyed) the manuscript, I only have a few comments, many related to clarity and rigor of equations and symbols. However, I would also like the authors to discuss with care the representativity of their results based on the choices made regarding configuration parameters (see comment on Line 191). All comments below:*

    We would like to thank Dr. Sole-Mari for his time reviewing our manuscript and the feedback offered to help us improve the rigor and clarity of our manuscript. We are glad you enjoyed reading our work! In general, we will address the problems with the notation in the equations and we will add clarifications to the model constraints and the derived analyses. Below, we address each comment in detail and outline specific revisions.

*Equation 1 and elsewhere: I think it is formally wrong to define K(h) and K(n) with the same exact symbol, which would seem to imply that they are the same exact function, and that here you just evaluate it for either h or n. You should use two different symbols for these two different factors.*

    The symbols $K(h)$ and $K(n)$ are replaced by $K_u(h)$ and $K_c(n)$, respectively, to indicate that they are different functions. The former refers to the change of hydraulic conductivity due to unsaturated conditions and the former refers to the change due to clogging.

*Equation 2b: I think you should point out at some point that the capillary head h is always negative, even if that might seem obvious to the authors, I think it is worth clarifying in order to ensure good interpretation of the equations with the minus signs in front of alpha h.*

    The following sentence with this clarification will be added to the manuscript.

    "Under unsaturated conditions, $h < 0$, whereas under saturated conditions, a positive pressure head replaces the matric head and $s_e = 1$"

*Equation 5: There seems to be a typo, the advection term should indicate the divergence of qCi (mind the dot as well as the parentheses).*

    The typo is corrected.

*Equations 5 and 6: Each species Ci should have a different reaction rate Ri, which is possibly a function of many other different "Cj". I believe you are wrongly using the parentheses (which should stand for "function of") as if they were part of the function's symbol. Brief: Ri, not R(Ci), and same for equations 6.*

    The notation to represent the reaction rates has been revised as suggested, and now reads $R_i$, where the subindex **i** refers to either organic carbon or dissolved oxygen.

*Line 100: X X*

    The typo has been corrected. This was meant to read $X_j$ which is the mass of biomass per unit volume. The subindex **j** refers to the different types of biomass considered in the model.

*Equations 9 and 10: Same comment as eq 5 and 6.*

    The notation has been replaced here as well. They now read $R_j$, where the subindex **j** refers to the biomass fractions.

*Lines 182-185: So you imposed a fixed gradient boundary condition at the inlet which changes over time in order to always get the 1mL/min inflow (or 0mL/min in dry conditions) ? So you kind of imposed a fixed flow at the inlet*

*boundary, or how is it different? Also, this reads "as shown in Figure 3", but Figure 3 doesn't really show much about how the boundary conditions are implemented. Maybe Figure 3 could indeed be improved to include more information.*

The fixed flow of 1mL/min is imposed until the system is clogged and ponding occurs. Since the hydraulic conductivity changes over time, the head gradient that needs to be imposed to maintain the fixed influx condition is recalculated at every timestep. From this calculation, the head value at the boundary is calculated. When the capillary head at the boundary becomes positive, it means that ponding is occurring. The fixed flow boundary condition is applied only until ponding occurs, after which the boundary condition becomes a constant head corresponding to the ponding in the experiment. We will add this explanation to the Methods section, and Figure 3 will be updated accordingly to better illustrate how the boundary conditions are set.

*Line 191: Why that choice of 450min? (which to me would seem like quite a short wetting time for operational realism). Later (section 3.5) you do seem to confirm this suspicion by finding quite a longer optimal cycle time, but at that point you have already fixed the dry-wet time ratio at 4.5. This leaves me wondering, for instance, if there isn't a more optimal strategy that uses rather long drying periods and also a lower dry-wet time ratio maximize hydraulic loading (unless I am missing something). I guess you could say that this is a first attempt and a framework which can be used for further investigation and optimization, but I think that some discussion around this possible limitation of the study is missing. In other words, you do discuss as of now that there are these two configuration parameters (wet-dry time ratio and dry time), but it should be made clear that there is probably a complex interplay between them and that for instance, for each different dry time, one may find different results regarding the role of the dry-wet time ratio.*

We chose that specific case just to demonstrate the effect of varying the $t_{dry}/t_{wet}$ while keeping $t_{wet}$ constant (Figure 6), and the effect of varying $t_{dry}$ while keeping the $t_{dry}/t_{wet}$ ratio constant (Figure 8).  We determined that this would be an interesting case to show the relation between biomass spatial distribution and hydraulic controls, as these trends were common among simulations. We ran the model for a wider range of $t_{wet}$ and $t_{dry}$ combinations and will add a Supplemental Figure showing the hydraulic efficiency results for these combinations. However, we did not intend to find a global optimum that is directly applicable to full-scale SAT applications. As you correctly point out, wet and dry periods are longer in full scale SAT systems. For the current SAT operation at Shafdan, wet periods are typically 1-2 days and dry periods are 2-4 days (Sharma & Kennedy, 2017; Idelovitch & Michail, 1984).  We will add discussion to the manuscript on the need for the model to be scaled up to optimize pilot and full-scale SAT systems.

*Line 295: I would say it as "Therefore, our results would suggest that neither...". Mostly because like I said earlier, you have not really simultaneously explored different ratios for different drying times.*

We ran the model for a wider range of $t_{wet}$ and $t_{wet}$ combinations and these findings were common among simulations. But we recognize that we make this statement based on our simulation, so it will be modified following your recommendation.

*Line 325: ,,*

The typo has been fixed.

*Sincerely,*
*Guillem Sole-Mari*

**References**

Idelovitch, E. and Michail, M.: Soil-Aquifer Treatment: A New Approach to an Old Method of Wastewater Reuse. Journal (Water Pollution Control Federation) , 56, No. 8, Conference Preview Issue (Aug., 1984), pp. 936-943, 1984.

Sharma, S. K. and Kennedy, M. D.: Soil aquifer treatment for wastewater treatment and reuse, International Biodeterioration & Biodegradation, 119, 671–677, https://doi.org/10.1016/j.ibiod.2016.09.013, 2017.

---

## Author Comment (AC2)

**Response to Comments – Referee #2**

*The manuscript "Continuum modeling of bioclogging of soil aquifer treatment systems segregating active and inactive biomass" by Saavedra Cifuentes et al. presents a numerical modeling study on the potential impact of dry and wet cycle iterations on the performance of soil aquifer treatment systems. The conceptual model approach considers different biomass pools (active cells, EPS, dead cells) each having its specific dynamics and contributing to changes of the hydraulic conductivity due to bioclogging. Data from a column experiment were used as a reference and subsequently the model was used to predict in influence of different configurations of dry and wet cycles on the exhibition of bioclogging and on the overall hydraulic performance of soil aquifer treatment systems.*

We thank the second referee for their detailed feedback. We will provide the requested clarifications in the revised manuscript by expanding the description of the column experiment of Rosenzweig (2011) and adding additional comparisons between the experimental results with our numerical simulations. In the original manuscript, we briefly summarized the experimental methods, focusing primarily on biomass distribution measurements, as they are most relevant to our conceptual and numerical model. To enhance clarity, we will now provide a more detailed account of these experimental methods in a supplementary section and include additional comparisons of simulation results with other measured variables from the experiment. Further information on specific changes and additions to the manuscript can be found in the responses to the reviewer's specific comments.

*General comments:*
*In general the manuscript is well written but exhibits some inconsistencies in the description of the used model approach – see specific comments below.*
*I acknowledge that the presented conceptual approach dividing the biomass into different pools is meaningful and consistent with other approaches presented in the literature. It is thus a valid hypothesis. However, I am not convinced how this hypothesis is tested/verified with results from the column experiment. There is indeed a match between the spatial distribution patterns on measured and simulated biomass but this comparison is done solely for the biomass and for a single observation time only.*

We appreciate the reviewer's comments. The available experimental data only covers the always-wet condition. Experimental results on various drying times and wet/dry ratios are not available.  Biomass profiles over depth are only available for a single time point at the end of the experiments because this requires destroying the sand column. We assess our numerical simulations with the available experimental data as an initial validation, which allows us to explore hypotheses related to drying cycles. Our simulations narrow down the range of optimal conditions, thereby guiding future experimental efforts. We acknowledge that reevaluation of the model will be necessary once new experimental data becomes available, particularly for larger-scale field applications. We will include this clarification in the manuscript.

*No comparison between measured and simulated substrate/electron acceptor concentration are shown and no comparison between measured and simulated changes of the flow dynamics have been shown either.*

In addition to biomass distribution, water content and matric head were recorded in the column experiment of Rosenzweig (2011), and we will add comparison of model simulations against these experimental results in the revised manuscript. We will add a more detailed description of the experimental measurements in the Methods section, along with a comparison of simulation results to these additional experimental observations in a supplementary section.

*Furthermore, I have the impression that the flux boundaries considered in the model are not adequate to describe the experimental conditions (or they are described in an insufficient way – see specific comments below). Therefore, other (less or equally complex) conceptual approaches might also explain the shown observations.*

We actually used a hybrid boundary condition with fixed flow until ponding occurs at the column inlet, and fixed head thereafter. We thank both reviewers for pointing out that our description of the BCs was incomplete. In the revised manuscript, we will clarify that the head gradient required to maintain this flux is recalculated at each time step as hydraulic conductivity decreases, and the boundary condition is switched to a constant head when the capillary head at the boundary becomes positive, indicating ponding. This clarification will be added to the Methods section, and Figure 3 will be updated to more clearly illustrate the boundary conditions.

*The same holds for the parameters used in this study. Most are taken from the literature but literature values for parameters describing microbial dynamics can easily vary by at least one order of magnitude and so does then vary the dynamics of the biomass itself. The presented results on the influence of the dry/wet cycle configuration might thus be biased. All this leads to the present study discussing the potential effects on clogging in SAT systems based on a reasonable but unverified hypothesis, only.*

We agree that microbial dynamics parameters can vary significantly across studies and applications. For the specific experiment evaluated here, the results presented in the paper show that our model provides a reasonable prediction of biomass distribution and clogging behavior with parameters from the literature. To ensure readers are aware of the limitations of available data, we will add discussion in the revised manuscript that further experimental data and parameter validation are necessary for field applications.

*I am not an expert for SAT systems but from their description I would assume that clogging at the bottom of the infiltration ponds is not only caused by microbial growth but also by the deposition of particles and organic material. How does this interfere with the discussed variations of the dry and wet cycle operation?*

You raise an important point regarding the potential for clogging in SAT systems being driven not only by microbial growth but also by the deposition of particles and organic material. Indeed, in the case of infiltration ponds, these factors are relevant and can influence clogging behavior. In the particular case of the Shafdan SAT, the water comes from the effluent of a secondary treatment thus particle concentrations are low (Idelovitch et al., 2003). Fluctuations in DOC and dissolved oxygen concentrations are also important and are influenced by photosynthesis in the ponded water (Goren et al., 2014). We will add information on all these additional complexities in the discussion section with recommendations on how these processes can be integrated in the modeling framework for specific field applications.

*Out of curiosity, I am also wondering if it is practically possible to tune the length of the wet cycle for a system affected by clogging – at the end a given amount of water is entering the pond and then infiltration takes as long as it takes.*

Yes, the length of wet and dry cycles in an SAT system can be adjusted based on the current system conditions and constraints. In practice, operators adapt the cycle lengths based on experience and limited real-time system monitoring (Sharma and Kennedy, 2017). The model we present provides a basis for improving operations via quantitative simulation and prediction. It can be applied adaptively with real-time or near-real-time ingestion of operating data.

*Similarly I am wondering if evaporation from the soil system during the dry cycles (especially in semi-arid regions) is interfering with the clogging effects, but I am aware that this is not the subject of this study.*

That's an interesting point. Evaporation could indeed affect the rate at which biomass desiccates during dry cycles, potentially influencing how quickly the system recovers to a near-clean state and how clogging is mitigated. For other applications, evaporation and desiccation rates could be incorporated into the model in future work. We will mention this consideration in

the Discussion section to acknowledge its potential impact on the system's behavior, especially in arid regions with high evaporation rates.

*Specific comments:*
*To my opinion the term "inactive biomass" is misleading since is it typically associated with biomass fractions (e.g., dormant cells or spores), which can turn into active biomass. In the present study the "inactive biomass" consists of EPS and dead cells. Some re-labeling of these pools might be helpful to avoid misunderstandings.*

The terms "active" and "inactive" biomass are standard in the literature, with "active" referring to biomass that can metabolize and replicate, and "inactive" referring to biomass that cannot, such as dead cells and EPS. These terms are widely used in environmental microbial modeling and biotechnologies (e.g., Rittmann & McCarty, 2020).

*Eq. 6b, l 104: Is this equation correct? If I insert the definition of $\zeta\_O2$ into Eq. 6b the term into the brackets and thus phase transfer rate approaches the non-zero negative value of $-s\_w*C\_02$ at saturation.*

Thanks for pointing this out. The last term in the parenthesis should not have been part of this equation. Instead, it should read:

$$\mathcal{R}_{O_2} = -\alpha_1\, r_{\text{DOC}|\textbf{ah}} X_{\textbf{ah}} + Mn\left(s_a C_{O_2|\text{sat}}\, \zeta_{O_2}\right)$$

With

$$\zeta_{O_2} = 1 - \frac{C_{O_2}}{C_{O_2|\text{sat}}}$$

This correction will be done to the manuscript.

*Eq. 7: I agree that pore-availaibility can limit microbial growth, but here it is assumed that microbial degradation activity is decreasing when the biomass is approaching the maximum volume. I.e., at highest biomass concentrations the activity is minimized. Some discussion on this would be needed.*

The term $\zeta_x$ in Equation 7 ensures that the source terms in Equation 10 are bounded to the pore-space available. Without this term, the biomass volume in the REV grows to values higher than the pore-space available in the REV. This clarification will be made in the manuscript. Importantly, lower microbial activity beyond the clogged layer is product of the accumulation of inactive biomass in the upstream pore space and the consequent restriction of substrate and nutrient fluxes. Our model is capable of simulating this coupling between pore fluid flow, microbial growth, and metabolism, at the REV scale.

*Fig. 2, Sections 2.2 and 2.3: In the Introduction and in Fig. 2 the model is introduced as considering different microbial processes incl. nitrification and denitrification but in Sections 2.2 and 2.3 only aerobic heterotrophs and their activity are described as part of the model. Clarify/correct.*

The application presented in this manuscript focuses exclusively on aerobic respiration, as it is expected to be the dominant process driving bioclogging in SAT systems. Figure 2 has been updated to reflect that only aerobic respiration is addressed in the presented study. We include discussion of the model's extensibility to other metabolic pathways such as nutrient transformations in the Discussion section, as suggested.

*Section 2.5: Was there any DOC present in the injected water or how was the carbon source applied? Are there any measured data on this which could be used for model verification? How long did the experiment last? Was there any measurement of the water fluxes at the effluent?*

Yes, DOC was present in the injected water, and we appreciate you highlighting this omission. The carbon source was lysogeny broth injected into the column at a 1:75 dilution, corresponding to a DOC concentration of 10 mg/L in the model. The experiment was run for 23 days. These details will be added to Section 2.5. A more detailed description of the column experiment will also be included in a supplemental section, with reference to Rosenzweig (2011) for additional data and results.

*L 177 adjust brackets around reference.*

The parentheses around the reference are fixed.

*L 178/179: Ok for the column experiment, but see also comments above on composition of the clogging layer. I guess the SAT optimization model is not describing a column experiment. Anyway, this discussion does not belong to this section.*

In those lines we describe the formation of the clogging layer as a result of our numerical model calculations, contrasting this approach with others in the literature where the clogging layer is treated as an input to a flow solver. We have kept this information in the Methods section, as it provides justification for the formulation of the numerical model.

*L 182: Figure 3 does not provide such information.*

Figure 3 will be updated to more accurately reflect the boundary conditions used in our numerical model.

*L 182-184: I do not get this. If the gradient is adjusted to maintain a given flux how does clogging result in a decrease of the simulated flux? To me it seems that constant flux conditions are simulated (at least during the wet periods) but later on the manuscript shows changes in the influx due to clogging are presented?!*

As we described in our previous response regarding the top boundary condition, the fixed flux condition is replaced with a constant head condition once ponding occurs, which accounts for the decrease in flux due to clogging.

*Tab. 2: Correct unit for "Half-reaction constant for electron acceptor"*

The typo will be corrected.

*Tab. 2: Why is a parameter for a nitrogen source given here? This is not mentioned in the equations given Sections 2.2 and 2.3.*

The table entry for the nitrogen source was removed.

*Tab. 2: The term "biodegradable fraction of dead biomass is misleading/incorrect". Both fractions of inactive biomass are decaying via hydrolysis.*

We agree that this terminology was confusing.  To provide consistent nomenclature, we will change the table entry to read "labile fraction of dead biomass".

*Tab. 2: Adding up the true yield and the fraction used for EPS gives a total fraction of 0.67 of DOC being converted into biomass. This sounds rather high. Comment/discuss.*

This is a typical value for aerobic heterotrophs when the DOC source is easily degradable (Rittmann and McCarty, 2020) and is reasonable for the conditions used in the experiment that we simulated. For applications to pilot- and full-scale SAT systems fed with wastewater effluent, we expect these values to be considerably lower.

*Units: The units used in Tabs. 1 and 2 are different than the units used in the text. While I understand the reasons for this, this is inconvenient for comparing text and table values.*

We will add unit conversions to Tables 1 and 2 to facilitate comparison with the text.

*L 187-189: Provide some further information on the biomass density. Is this given as dry mass per wet volume or carbon mass per volume or something else. Since the yields are dimensionless I guess it is DOC mass per volume of biomass. Assuming the latter the presented value is in agreement with literature values for bulk biomass (i.e. bacterial cells plus EPS etc.) and the rather low value is explained by low density EPS forming much of the bulk biomass. To use this value for the active bacterial cells is no well justified since the carbon content of cells is much higher and thus their density also much higher. This implies that with the used density the volume and thus the clogging effect of the active and recalcitrant biomass (living and dead bacterial cells) is overestimated.*

The term $\rho_X$ represents the biomass density with units of dry mass per wet volume. Under the macroscopic scale of our model, it should be interpreted simply as a scaling factor between biomass content and the reduction in hydraulic conductivity (Kildsgaard and Engesgaard, 2001). $\rho_X$ is the most sensitive parameter in bioclogging simulations and it is usually used as a fitting parameter. The suggestion of keeping separate $\rho_X$ terms for active and inactive biomass is valid but it just introduces an additional parameter that is not supported by independent measurements, so including it would not add new information to the model. Other values reported in the literature for $\rho_X$ are 1 kg/m$^3$ (Caruso et al., 2017), 2.5 kg/m$^3$ (Clement et al., 1996), 5 kg/m$^3$ (Kildsgaard and Engesgaard, 2001), and 17.5 kg/m³ to 50 kg/m³ (Mostafa and Van Geel, 2012). We appreciate this discussion, and we will include these considerations in the manuscript.

*Fig. 4: Compare for which cell mass the two curves would match. From Fig. 2 I get the impression that approx. 10^-11 g/cell are considered. Are these values reasonable?*

The CFU count at the topmost section was $3.5 \times 10^9$ CFU/mL which is equivalent to $3.75 \times 10^{12}$ CFU/kg dry sand. Considering that a single bacterial cell weights around $10^{-12}$ g (Madigan et al., 2021), this equates to 3.8 g active biomass/kg dry sand or 5 kg/m³, which compared to the simulation results, we consider to be a reasonable result, given the limitations of the CFU method.

*L 200-202: The statement is reasonable but again, are there any measured concentrations from the experiment?*

DOC was measured at the outflow at the end of the experiment, which indicated that 89% of the substrate was consumed.

*Fig. 5a and c: If you want to emphasize that the flux is going down invert the y-axis (or show the absolute value of the flux). Otherwise you rather leave the impression of an increasing flux.*

Thank you for the suggestion. The y-axis in Figure 5 will be flipped to avoid any confusion regarding the decreasing fluxes.

*Btw., are the no data on measured water fluxes from the experiment?*

Water flux measurements were taken daily from both the control and the inoculated columns. Further, matric head and water content data were collected over time and over depth on the topmost 0.30m of the column. We will add this information to the revised manuscript, compare observations to the results from the numerical simulations, and provide the experimental data in Supplemental Figures.

*Fig. 5: Clarify which y-axis values have to be multiplied by which factor. The "1e-6" seems to be out of place and the "x10^-2" might belong to panels a/c or b/d.*

The "1e-6" and "x10^-2" notes have been removed and replaced with notation directly on the y-axis ticks to eliminate confusion.

*L 227/Figure 5c: I do not see specific dips in the influx which I could attribute to the dry periods. It appears as if the influxes gradually decrease which is not what I would expect for a dry period.*

Thank you for pointing this out. The y-axis of Figure 5c was mislabeled; it should read "outflux from the column." We have corrected the figure labels and descriptions accordingly. We will add a supplementary figure showing both influx and outflux to clarify this point and better illustrate the behavior during dry periods.

*L 232: There is no consumption of inactive biomass by the active cells described by the equations in Sections 2.2 and 2.3.*

The hydrolysis of inactive biomass yields DOC, which contributes to the source terms in Equation 6a. To avoid confusion, we have clarified this in the manuscript and updated Figure 2 to reflect this information.

*L 235-236: Unclear/rephrase. Is the mentioned hydraulic loading the average loading rate at quasi-steady state or including the initial phases, too? From Fig. 5 I get the impression that 2.3x10^-2 m/s is rather the initial value and not the long term value.*

The long-term hydraulic loading rate is defined in Section 2.7 as the total infiltrated volume per unit area divided by the duration of the experiment. The value of 2.3x10-2 m/s mentioned here corresponds to this definition. However, this value is not shown in Figure 5 where flux over time is plotted; the referee may have been referring to Figure 6 instead.

*L 254-256: Clarify. From this sentence I is not clear to me what you define as long-term hydraulic loading rate. Clarify also if the presented values are averages for a full wet/dry cycle (or if they are something else).*

The long-term hydraulic loading rate is defined in the Methods section as the total infiltrated volume per unit area divided by the duration of the experiment. We will reinforce the definition at this point in the revised manuscript for clarity.

*L 264 and below: Here it would be highly interesting if the shown biomass distributions are at or close to steady state (I guess not) or if they would approach a different distribution at later times. Since the clogging effects are mainly caused by the high biomass concentration at the vicinity of the inflow a steady state of the fluxes does not necessarily imply a steady state of the biomass in the downstream regions.?*

The simulation duration was sufficient to reach an apparent steady state. For biomass distributions, we tested for a steady state by integrating the biomass fractions over depth and ensuring that they did not change significantly between cycles. However, as the reviewer correctly points out, this only reflects a steady state in the upper regions, as total biomass is primarily driven by accumulation in the topmost layer. Thus, a complete steady state may not have been achieved, particularly in the deeper regions of the column.

*It would also be good to show some substrate concentration results as they would indicate if growth would be possible in the deeper regions of the columns. For the interpretation of the results and for the potential implications for real SAT systems one would also need to know what is limiting microbial activity: depletion of DOC or of O2*

Our simulations indicate that DOC was always consumed before dissolved oxygen, likely due to the unsaturated conditions in the column, which allowed oxygen to be constantly replenished in the water phase. To better illustrate these conditions, we will include depth-time heatmaps for dissolved oxygen and DOC. The model will need to be extended for larger scale application to full-scale SAT systems, which involve additional complexities. We will add discussion of the extension to larger scale in the Discussion section.

*L 282-284: I do not get this statement.*

The statement explains the trends in Figure 8b. The hydraulic loading rate increases with longer drying times because it reflects the extent to which bioclogging is reversed, allowing the soil to recover its infiltration capacity.

*Section 3.5: Similarly to my comment above, I would be good to know if the presented biomass concentrations are at steady state or at least close to it.*

Yes, the system is at a steady state, at least for the upper part of the column. We will add a note in Section 3.5 to indicate this.

*L 331-332: Do you have any additional results confirming this statement or is this based on the biomass distribution data only? If the later is the case, I think this statement is misleading.*

This statement is based on the biomass distribution and water content profiles, which are the data available from the column experiments.

*L 340-341: This is the first time some information on substrate supply in the column experiment is provided. This information and further details should be added to Section 2. Which DOC concentrations was provided in the column experiment?*

Information on substrate supply in the column experiment will be added to the Methods (Section 2) for clarity, and additional detail on the experiments performed by Rosenzweig (2011) will be included as Supplemental Information.

**References**

Caruso, A., Boano, F., Ridolfi, L., Chopp, D. L., & Packman, A.: Biofilm-induced bioclogging produces sharp interfaces in hyporheic flow, redox conditions, and microbial community structure. Geophysical Research Letters (Vol. 44, Issue 10, pp. 4917–4925). https://doi.org/10.1002/2017gl073651, 2017.

Clement, T. P., Hooker, B. S., and Skeen, R. S.: Macroscopic Models for Predicting Changes in Saturated Porous Media Properties Caused by Microbial Growth, Groundwater, 34, 934–942, https://doi.org/10.1111/j.1745-6584.1996.tb02088.x, 1996.

Goren, O., Burg, A., Gavrieli, I., Negev, I., Guttman, J., Kraitzer, T., Kloppmann, W., and Lazar, B.: Biogeochemical processes in infiltration basins and their impact on the recharging effluent, the soil aquifer treatment (SAT) system of the Shafdan plant, Israel, Applied Geochemistry, 48, 58–69, https://doi.org/10.1016/j.apgeochem.2014.06.017, 2014.

Idelovitch, E. and Michail, M.: Soil-Aquifer Treatment: A New Approach to an Old Method of Wastewater Reuse. Journal (Water Pollution Control Federation) , 56, No. 8, Conference Preview Issue (Aug., 1984), pp. 936-943, 1984.

Idelovitch, E., Icekson-Tal, N. , Avraham, O., Michail, M.: The long-term performance of Soil Aquifer Treatment (SAT) for effluent reuse. *Water Supply,* August 2003; 3 (4): 239–246. https://doi.org/10.2166/ws.2003.0068, 2003.

Kildsgaard, J., & Engesgaard, P. Numerical analysis of biological clogging in two-dimensional sand box experiments. In Journal of Contaminant Hydrology (Vol. 50, Issues 3–4, pp. 261–285). Elsevier BV. https://doi.org/10.1016/s0169-7722(01)00109-7 , 2001.

Madigan, M. T., Bender, K.S., Buckley D.H., Sattley, W. M., Stahl, D.A., Brock Biology of Microorganisms, 16th edition. Published by Pearson (July 1, 2020) 2021.

Mostafa, M. and Van Geel, P.: Validation of a Relative Permeability Model for Bioclogging in Unsaturated Soils, Vadose Zone Journal, 11,vzj 2011.0044, https://doi.org/10.2136/vzj2011.0044, 2012.

Rosenzweig, R.: The effect of biofilms on the hydraulic properties of unsaturated soils, Ph.D. thesis, Technion-Israel Institute of Technology, 485 Haifa, Israel, 2011.

Rittmann, B. E. and MacCarty, P. L.: Environmental biotechnology: principles and applications, McGraw-Hill, New York, second edition, 2020.

Sharma, S. K. and Kennedy, M. D.: Soil aquifer treatment for wastewater treatment and reuse, International Biodeterioration & Biodegradation, 119, 671–677, https://doi.org/10.1016/j.ibiod.2016.09.013, 2017.

---

## Author Response (AR2)

In this document we indicate where changes have been to the manuscript following the comments made by the referees. Line numbers refer to the tracking changes document version. A short description of the changes made is emphasized with **blue text**.

—————

**Response to Comments and Edits to Manuscript**

*Dear Authors,*

*The revised version of the manuscript addresses appropriately many of the comments raised by the two Reviewers. However, Reviewer #1 still raises some points concerning mainly i) the lack of information on how well the model is simulating substrate consumption for the reference experiment; ii) the need to compare modelled and measured water fluxes and/or hydraulic heads; iii) the need to clarify the "additional comparisons between the experimental results with our numerical simulations".*

*I recommend "Moderate Revision" and asks the Authors to respond to all points raised by Reviewer #1.*

*Kind regards,*

*Roger Moussa*

**Overall Response**

Dear Dr. Moussa,

Thank you for handling our manuscript and for highlighting the areas where further clarification is needed. We also appreciate the valuable comments from the Reviewers, which have helped us improve the quality of our submission. At the core of this work is our mathematical model, which couples three key components: unsaturated flow, nutrient consumption and fate and transport, and biomass growth. The core novelty lies in the coupling of these three components with the explicit calculation of inactive and active biomass, and resulting impacts on performance of soil aquifer treatment systems. This advance provides a more rigorous representation of the biological processes occurring within the system that is critical to system clogging. This new representation is the reason why we must use the unique biomass characterization data obtained from this particular column experiment, as such detailed experimental data for both active and inactive biomass is rare.

We acknowledge the experimental data are not specific to the model parametrization, in fact, these experiments were performed in 2011 without this particular modeling effort in sight. Beyond the contributions to simulating clogging processes, our manuscript aims to motivate the development of experimental methods that separate active and inactive biomass in porous media, as our results show that this is an important process but generally has not been considered in either experimental or modeling studies to date.

To address your and Reviewer #1's concerns regarding the substrate consumption and unsaturated flow components of the model, we have added model-data comparisons and clarified the limitations of the experimental data.

**Detailed Response**

- Regarding **point i)** concerning substrate consumption: Samples from the column feed solution and effluent were taken only at the experiment's conclusion. These measurements showed 89% removal of the substrate, which is comparable to the 99% removal of DOC predicted by our model at the same time point.

- Regarding **point ii)** about unsaturated flow: We have added a comparison between the modeled and measured water content profiles. New plots are included comparing the water content sensor data and simulation results over time at various depths.
- Finally, **point iii)** regarding additional comparisons between experimental results and numerical simulations: As requested by Referee #2, we have made several additions throughout the revised manuscript to highlight and clarify all comparisons made between the experimental results and our numerical simulations. The specific changes addressing these comparisons are detailed in our individual response to Referee #2, which accompanies this submission.

We want to reiterate that while we have addressed the concerns regarding the substrate consumption and unsaturated flow components, our primary focus in this manuscript remains on presenting and validating the novel aspect of our model related to the explicit treatment and calculation of active and inactive biomass, and its key role in bioclogging and implications for soil aquifer treatment systems, which are emerging as a critical approach for wastewater reuse. We hope these revisions fully address your remaining points. These revisions have strengthened the manuscript, and we appreciate the time and effort invested in evaluating our work.
* * *
**Referee #1**

*Nothing to change.*

We thank the reviewer for his positive evaluation of the revised version of our paper.
* * *
**Referee #2**

*2.1) The revised version of the manuscript "Continuum modeling of bioclogging of soil aquifer systems segregating active and inactive biomass" by Cifuentes et al. (hess-2024-251) addresses many of my original comments appropriately. There are however some points where I am not satisfied by the replies/revisions of the authors. My main concern is that there is still no information on how well the model is simulating substrate (and electron donor) consumption for the reference experiment. To my opinion, this is one of the core criteria for the accuracy of a model applied to a biodegradation setting and I am somewhat surprised that I need to argue for that.*

Samples from the column feeding solution and effluents were taken only at the end of the experiment for total organic carbon analysis and that analysis showed 89% removal of the substrate. In the simulations, 99% of the input DOC is consumed. There was no continuous monitoring of effluent substrate during the experiment, but this end-point measurement provides a reasonable check, especially considering SAT systems generally lead to near-complete DOC consumption. **We added Lines 226 to 231 to the Results section addressing this comparison and acknowledging the limitations of the data. We also added a plot of simulated DOC concentration over time as Supplemental Figure A16.**

*Similarly, I do not find a comparison between modeled and measured water fluxes and/or hydraulic heads. The only data shown as indication for the accuracy of the model simulations are the biomass data in Fig. 3 and the water contents in Fig. A13. In their reply the authors claim to have added "additional comparisons between the experimental results with our numerical simulations" but I can not identify and futher comparisions than those mentioned above. This challenges consequently all conclusions drawn from the model scenarios presented in the manuscript.*

**We added an additional comparison between water content sensor data with the simulation results as Supplemental Figure A14.** This complements the end-state comparison presented before between simulated and water content. The agreement is reasonable for the upper layers of the column but it is poorer on the deeper layers. We argue that the model's ability to capture the general trends of water content fluctuations is sufficient to say that it provides a fair representation of the internal hydraulic conditions in the system. We considered that the tensiometer sensor data and the simulated

matric head are not readily comparable because it is expected that unsaturated flow parameters (i.e., the van Genuchten-Mualem model parameters) will change as biomass accumulates. Our proposed model does not capture or solve for such changes and it is one of the simplifications we made in the model conception. **We added Lines 86 to 88 in the Methods section to detail this consideration in the proposed model.**

*Some further minor points (comment number refer to the reply list provided by the authors):*

*Comment 2.13: Clarify, in the reply you mention DOC concentrations of 10 mg/L in the manuscript it is 100 mg/L. Also, are such concentrations comparable to the values considered for SAT systems?*

DOC concentrations on a SAT system are lower than the concentration used in the experiment. The DOC source in a SAT system is the effluent of a wastewater treatment plant and those concentrations are around 15-20 mg/L (Idelovitch et al., 2003). In the column experiment, the DOC source is a growth broth which is characterized with a concentration of 100 mg/L in the simulations. We actually ran simulations with other influent concentrations and found that the system reached the same end-state: accumulation of biomass near the top clogging the system. In terms of experimentation, running a column with actual wastewater treatment plant effluent would be interesting because DOC bio-availability is much lower than that of the growth broth. This kind of column experiments are rare and a note for future directions with these considerations is added in the manuscript. *To clarify this point, we added Lines 392 to 398 to the Discussion section.*

*Comments 2.21: If the yields are expected to be "considerably lower" in the SAT systems that in the column experiments, what does this mean for extrapolating the results to the SAT systems. Less clogging, slower clogging or …? Here and at for some other comments I think the manuscript would benefit if arguments/clarifications/etc. are not only given in the reply but also find their way into the manuscript.*

Relative to the column experiments, lower yields are expected in SAT systems primarily due to the nature of the dissolved organic carbon (DOC) typically present in secondary-treated wastewater effluent. This DOC is generally more refractory and less readily metabolized by microorganisms compared to the growth substrate used in the controlled column experiments. This difference in substrate bio-availability would not necessarily lead to less clogging in SAT systems in the long term, but rather to slower clogging initially. The accumulation of refractory fraction of inactive biomass that we found to be a key driver of bioclogging will still occur: accumulation of recalcitrant material will eventually dominate the clogging process, leading to the clogged end state we observe and simulate in our model. *We have added this information to the Discussion section of the manuscript in Lines 398 to 404.*

*Comments 2.23: If rho_x is the most sensitive parameter in clogging simulations (which I agree on) than one should argue carefully if the presented value is meaningful. The low value for the density provided here is similar to values other authors have used for "bulk biomass". If the bulk biomass mainly consists of EPS such low values can be justified. If the bulk biomass mainly consists of bacterial cells this is not possible. When comparing result with the literature it is thus necessary to be more careful than what is done here.*

While bulk biomass ($\rho_x$) has a physical basis in Equation 11, its treatment as a calibrated parameter can obscure this significance. The composition of bulk biomass in the simulation changes over time: at very early times, the bulk biomass is dominated by the active fraction, but inactive biomass accumulates in the porous medium and becomes the dominant biomass fraction later. Therefore, the accumulation of inactive biomass means that lighter-weight EPS become a larger fraction of the total biomass over time, and the cell density within the biomass correspondingly decreases over time. *We added this insight into the manuscript in Lines 368 to 373 in the Discussion section.*

*Comment 2.25: In addition to my comments above: What is the corresponding value predicted by the model?? There should be changes made to the manuscript!!*

The value of $\rho_x$ used was mentioned briefly in Section 2.6 of the original manuscript. *For clarity, we added it to Table 2 of the revised manuscript to make the information easier for readers to find.*

*Comment 2.27: I do not find the mentioned comparison between observations and numerical results.*

To clarify this point, *we added plots (Supplemental Figure A14) that explicitly compare the water content sensor data with the simulation results.* These new plots complement the end-state water content comparison presented previously.

*Comments 2.32: I acknowledge that the definition is take from the literature. However, I think one should spend some words on the fact that the relative order of the values for the different scenarios depends on the (arbitrary length) of the experimental length. At the end the initial behavior of the systems differs from the behavior at later times.*

Thanks for the suggestion. Yes, a characteristic time scale was needed for the definition of long-term hydraulic loading. We chose the length of the experiment for this, as it accounts for both the initial high flows into the clean column and the later low flows after clogging occurs. For SAT applications, this time scale should be chosen to span system resets, meaning it should cover the dry periods when there is no water inflow and until the system infiltration capacity is restored either by a much longer drying time or some mechanical intervention. **We added this consideration to the Discussion section in Lines 347 to 351.**

**References**

Idelovitch, E., Icekson-Tal, N. , Avraham, O., Michail, M.: The long-term performance of Soil Aquifer Treatment (SAT) for effluent reuse. *Water Supply,* August 2003; 3 (4): 239–246. https://doi.org/10.2166/ws.2003.0068, 2003.